# CHESSFORMER: A UNIFIED ARCHITECTURE FOR CHESS MODELING

**Daniel Monroe**[1]    **George Eilender**[1]    **Philip Chalmers**[2]    **Zhenwei Tang**[1]    **Ashton Anderson**[1]
[1]University of Toronto    [2]Williams College

## ABSTRACT

Chess has long served as a canonical testbed for artificial intelligence, but modeling approaches for its central tasks have diverged. Maximizing playing strength, predicting human play, and enabling interpretability are typically solved with disparate architectures, and these designs are often misaligned with the geometry of the domain. This raises the natural question of whether these objectives require separate modeling paradigms, or if there exists a single architecture that supports them simultaneously.

We introduce Chessformer, a unified architecture that advances the state of the art on all three central goals in chess modeling. Chessformer is an encoder-only transformer that represents board squares as tokens, augments self-attention with a novel dynamic positional encoding called Geometric Attention Bias (GAB) that adapts to domain-specific geometry, and predicts actions with an attention-based source-destination policy head. We evaluate Chessformer on each front. First, we develop MAIA-3, a family of models for human move prediction that reaches 57.1% move-matching accuracy, significantly surpassing the previous state of the art with fewer than a quarter of the parameters. Second, we integrate Chessformer into Leela Chess Zero, a leading open-source engine, adding over 100 Elo of playing strength and resulting in tournament victories over Stockfish in major computer chess competitions. Third, we show that Chessformer's square-token design makes attention patterns and activations directly attributable to board squares, enabling granular interpretability analyses that prior architectures do not naturally support. More broadly, our results demonstrate that aligning a model's tokenization, positional encoding, and output design with the underlying structure of a domain can yield simultaneous gains in performance, human compatibility, and interpretability.

## 1 INTRODUCTION

A central goal of artificial intelligence (AI) is to build systems that are simultaneously high-performing and human-compatible. Models that are both intelligent and intelligible promise to not only be powerful on their own, but to also augment human partners through collaboration, teaching, and mutual understanding. Chess is a particularly well-suited model system for this dual goal: while modern chess engines are decisively superhuman, their behavior often diverges significantly from human players and remains opaque even to experts.

However, the current chess modeling literature is fragmented. Systems for raw strength include alpha-beta search engines, self-play policy-value networks with MCTS, and transformers with linearized board representations (Campbell et al., 2002; Romstad et al., 2023; Silver et al., 2018; Ruoss et al., 2024), while human move-prediction systems range from convolutional stacks over board images (McIlroy-Young et al., 2020; Tang et al., 2024) to autoregressive models of move histories (Zhang et al., 2025). This raises a natural question: can one architecture support strength, human modeling, and interpretability simultaneously?

We introduce Chessformer, a unified architecture that advances the state of the art on three fronts at once: it produces substantial gains over prior methods in raw chess-playing ability, it surpasses state-of-the-art human move-matching performance with a fraction of the parameter count, and

it admits downstream interpretability analyses more naturally than previous models. Concretely, Chessformer is an encoder-only transformer that treats the 64 board squares as tokens, pairs this square-token body with an attention-based "source-destination" policy head, and equips the trunk with Geometric Attention Bias (GAB), a novel dynamic positional-bias generator that adapts attention to the geometry of a chess position.

We demonstrate Chessformer's performance on all three objectives. First, we report on integrating the Chessformer design into Leela Chess Zero, a leading open-source chess engine. Chessformer increased the playing strength of Leela by over 100 Elo points (for reference, major chess engine releases rarely surpass previous versions by more than 50 Elo points (Stockfish Team, 2026)), which resulted in configurations that defeated the perennial world champion Stockfish in elite computer chess tournaments. Second, we use the Chessformer architecture to train MAIA-3, a series of models that represent the new state-of-the-art in human chess move prediction. Our 79M-parameter model achieves a move-matching accuracy of 57.1%, surpassing the previous best of 55.9%, which was attained by a prior method's 355M-parameter model with search enabled (Zhang et al., 2025). Third, we present interpretability findings that are enabled by Chessformer's domain-aligned architecture. Chessformer learns many fine-grained and interpretable features corresponding both to well-known chess concepts, such as forks and pins, and to lesser-known patterns that capture meaningful variation.

Empirically, we find that GAB is a key driver of Chessformer's generalist abilities. Our ablations show it contributes significant improvements over absolute and relative position encodings for Elo, puzzle accuracy, and policy and value accuracy, with sizable performance-per-compute efficiency gains. These results reinforce a broader lesson: adapting model architecture to a domain's structure allows AI models to more flexibly adapt to both task mastery and human compatibility. In chess, this yields a single model family that improves engine strength, advances human move-matching, and enables square-level interpretability, all while being substantially more efficient than previous approaches. We open-source all code and data at `https://github.com/CSSLab/maia3`.

## 2 RELATED WORK

Computational approaches to chess have traditionally focused on maximizing absolute playing strength by developing hand-crafted search heuristics and position evaluations. This approach gave rise to Deep Blue (Campbell et al., 2002), which in 1997 became the first computer to defeat a reigning human World Chess Champion in a match, and Stockfish (Romstad et al., 2023), which is generally considered the strongest chess engine available today. AlphaZero (Silver et al., 2018) introduced a different recipe based on Monte Carlo tree search (MCTS) and reinforcement learning, training neural networks through self-play to predict state values and policy distributions over subsequent actions. The open-source re-implementation Leela Chess Zero (Pascutto & Linscott, 2019) refined this approach with new neural network architectures and search strategies and often ranks as a close runner-up to Stockfish in computer chess competitions. Even without search, transformer-based agents can achieve grandmaster-level strength when strong oracles are distilled into them (Ruoss et al., 2024).

A more recent line of work aims to develop chess systems that are not only strong but also human-compatible, in the sense that they understand and are attuned to the behavior of humans, by modeling human play across skill levels. This was first explored by MAIA (McIlroy-Young et al., 2020), which employed a set of convolutional neural networks, each trained to model players at a specific rating range. Jacob et al. (2022) showed how to adapt MCTS to produce slightly better move-matching performance. MAIA-2 (Tang et al., 2024) simplified MAIA's approach with a unified model, introducing a skill-aware self-attention layer that tokenized the channels of the output of a stack of convolution layers. ALLIE (Zhang et al., 2025) viewed this behavior replication task through the lens of language modeling, training a decoder-only transformer model on a move-based representation of the game trajectory to achieve state-of-the-art human move-matching. These human emulation methods were leveraged by Hamade et al. (2024) to investigate human-AI cooperation in chess. McIlroy-Young et al. (2021) demonstrated that individual human players can be reliably identified from just a few of their games, while McIlroy-Young et al. (2022) improved move-matching performance on individual players by finetuning on their games. Most recently, Tang et al. (2025) introduced MAIA4ALL, showing that individual-level human move prediction

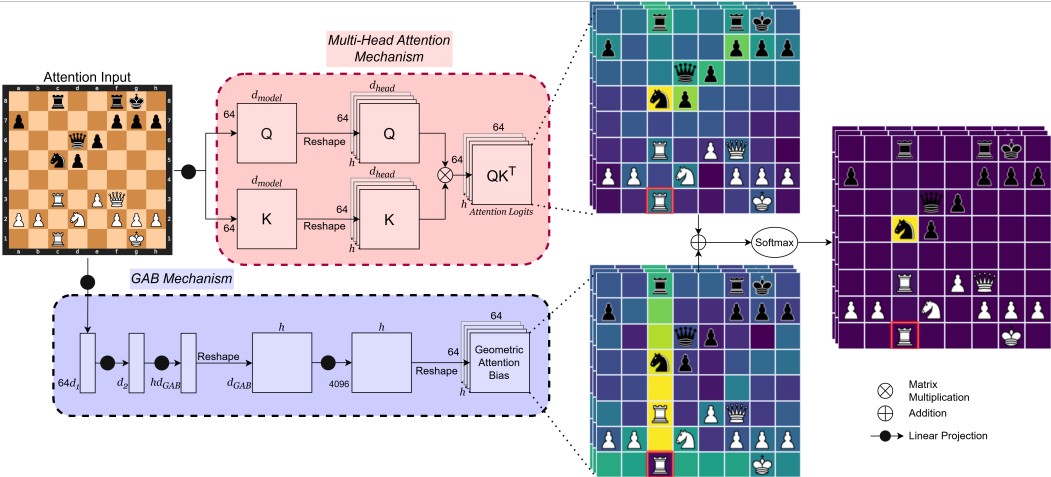

Figure 1: Chessformer attention mechanism. Chessformer adopts the natural visual representation of the chessboard, processing the 64 board squares as tokens. It augments the dot-product attention mechanism with *Geometric Attention Bias* (GAB), a novel position encoding that generates biases for attention logits from a compressed representation of the board state. A consistent theme of attention in Chessformer models is harmonious collaboration between the semantic dot-product attention component and the positional GAB component. In the example shown, the querying square is white's rook on c1, highlighted in red. Higher attention scores are colored yellow, while lower attention scores are colored purple. The dot-product attention logits highlight important pieces, while the GAB biases focus on squares that are a rook's move away. Together, they point white's rook to the pinned Black knight on c5.

can be made substantially more data-efficient by using prototype players and a meta-network to initialize player-specific embeddings from only a small number of games.

Chess has also served as a model system for other problems in AI. Farebrother et al. (2024) used chess to demonstrate that classifying rather than regressing improves scalability in deep reinforcement learning, while Feng et al. (2025) investigated creative generative AI using chess puzzles as a testbed. In mechanistic interpretability, which studies the mechanisms underlying the behavior of AI models, McGrath et al. (2022) used linear probes to identify concepts learned by AlphaZero, while Jenner et al. (2024) found evidence of learned planning in a transformer model trained by the Leela Chess Zero project.

## 3 METHODOLOGY

We now describe our Chessformer architecture and training setup. Chessformer is an encoder-only transformer that processes the 64 chessboard squares as tokens and augments self-attention layers with a novel dynamic position encoding called Geometric Attention Bias (GAB). We evaluate this architecture on three objectives in chess modeling: emulating human play, maximizing raw playing strength, and enabling interpretability analyses. For the first two tasks, we assemble a dataset of chess games and train Chessformer models in a supervised fashion to take in board states from those games and predict both game outcomes and played moves (or for playing strength, distributions over moves corresponding to playouts chosen during engine self-play).

### 3.1 BOARD REPRESENTATION: SQUARES AS TOKENS

Existing transformer-based methods in chess use a variety of tokenization schemes: ALLIE represents a position via the sequence moves taken to reach it, MAIA-2 tokenizes the channels of the output of a stack of convolution blocks, and Ruoss et al. (2024) applies rotary position embeddings (Su et al., 2024) on top of a tokenized representation of the 64 board squares and additional information contained in the Forsyth-Edwards Notation (FEN) (Edwards, 1994) representation of

the position. We argue in Appendix D that these formulations are misaligned with the geometry of chess as a domain and may therefore hinder performance. Chessformer instead adopts the natural visual representation that treats individual chessboard squares as tokens, creating fixed positional relationships between tokens that are grounded in the domain and can be captured effectively by position encodings. It also allows tokens to specialize to their corresponding squares rather than represent the entire board state, which greatly reduces the load on parameters.

Concretely, positions are represented as a sequence of 64 one-hot or zero vectors of dimension 12, each indicating which of the 12 pieces are present on the corresponding square, and the board is flipped to the perspective of the side to move. To obtain the input for a given board state, we concatenate representations of the current and past $n$ positions, where $n$ is a non-negative integer controlling the amount of history information conditioned upon, and we repeat the earliest position if some or all of these past positions are not available. Unless otherwise stated, $n = 7$. For the engine distillation setup, we also concatenate auxiliary information that was necessary for compatibility with the Leela Chess Zero infrastructure, though this did not noticeably affect performance in initial experiments (see Appendix A.2). The final board representation consists of 64 tokens, one for each square.

Our setup is most similar to that of Ruoss et al. (2024), which tokenized the 64 board squares in addition to other information about the position, for a total of 77 tokens. However, that work adopted rotary position embeddings (Su et al., 2024) on a linearized representation of the board squares, enforcing a one-dimensional structure on the tokens that is not grounded in the domain. This is especially deleterious given the central role that position plays in chess. For example, among relationships between squares, their architecture maximally decays the attention strength between opposite corners, even though those corners lie on a main diagonal that is critical for certain pieces and tactical patterns.

## 3.2 GEOMETRIC ATTENTION BIAS

Self-attention in transformer architectures is permutation-invariant, so positional information must be introduced to the model through some kind of position encoding. In language and vision settings, simple Euclidean distance is arguably the predominant notion of position, and thus static position encoding schemes, like rotary position embeddings and absolute and relative biases, power state-of-the-art models in these domains (see Appendix C). However, chess follows its own special geometry, in which the six piece types move in particular ways. In addition, positional relationships can vary widely with the board state in chess. As a simple example, relationships corresponding to the movement of a particular piece are only sensible if that piece is present on the board. But chess is rife with more complex interactions; for example, connections between distant squares are weaker in closed positions (those with fixed pawn structures). A more versatile positional encoding is necessary.

To capture the variable geometry of chess, we propose an adaptive position encoding called Geometric Attention Bias (GAB). GAB uses a compressed representation of the board state to generate biases for each attention head from a set of templates. To compress the board state, tokens first undergo a linear projection of dimension $d_1$ and are flattened, followed by a linear projection of dimension $d_2$ with GELU activation and layer normalization. To generate the attention biases, we apply another linear projection of depth $h \cdot d_3$ followed by activation and normalization, and reshape to $h \times d_3$. We apply a final linear projection, shared by the whole model to reduce parameter count and accelerate learning, to form biases of shape $h \times 4096$ which are reshaped to $h \times 64 \times 64$ and added to the dot-product logits before softmax. This final projection can be viewed as dynamically mixing a set of $d_3$ attention bias templates. Figure 4 presents pseudocode for generating GAB biases.

Our approach has a number of benefits. First, it models positional information globally rather than through individual tokens. This aligns well with our later finding that Chessformer models mainly adapt the GAB component of the self-attention computation to global positional features like the game stage (opening, middlegame, endgame), rather than local features like the locations of individual pieces. Second, representing attention logits as the sum of a semantic component generated by dot-product attention and a positional component generated through GAB allows self-attention layers to assign relevant aspects of the attention computation to each part. The choice of a dynamic position encoding enables attention heads to be repurposed based on the board state, a

behavior we explore in Section 6. Finally, formulating the interaction additively allows us to use existing memory-efficient attention kernels during training and inference (Dao et al., 2022).

### 3.3 OUTPUT HEADS

All models we train have two output heads: a value head that predicts the game outcome (*win*, *draw*, and *loss*), and a policy head that predicts the move (or in the case of engine oracle self-play games, the distribution of moves corresponding to playouts) chosen during the game. To generate the game outcome prediction, we apply mean pooling to the encoder body's output, followed by layer normalization. We then apply a linear projection to dimension 128, followed by a ReLU nonlinearity, followed by a linear projection to the three logits for the game outcome prediction target.

Prior work has modeled move distributions in a variety of ways. ALLIE autoregressively predicted tokens corresponding to each of the 1968 possible moves in Universal Chess Interface (UCI) notation, while MAIA-2 used an MLP layer. We propose a policy head based on self-attention that reflects the "from-to" structure of the underlying action space, encoding moves by the starting square and destination square of the moved piece. Given the 64 tokens returned by the encoder body, we generate via linear projection a set of query vectors corresponding to the starting square and a set of key vectors corresponding to the destination square, both with depth equal to the depth of the encoder body. Logits for moves are calculated via scaled dot-product, resulting in a $64 \times 64$ matrix representing all possible traversals from one chessboard square to another. This is sufficient to represent all moves except promotions, implementation details for which are reported in Appendix A.3.

The from-to formulation matches the action space's structure and substantially improves interpretability (see Section 6) without sacrificing performance. We hope that this design choice will catalyze future mechanistic interpretability research on Chessformer models.

## 4 PREDICTING HUMAN PLAY: MAIA-3

We first apply Chessformer to the problem of predicting moves from real human games. Given a board position encountered in a game, and the ratings of the players who played the game, our objective is to predict the move that was actually played in the game. A model that does this well has many useful applications, including human-like gameplay, modeling the trajectories that players take as they improve, capturing natural mistakes, providing useful input to teaching and tutoring systems, and so on.

### 4.1 DATASET

We construct a training dataset consisting of blitz games played on the online chess platform Lichess from January 2023 to July 2025. As the bulk of these games are from the middle of the skill distribution, we re-sample the games during training so that all skill levels are equally represented. The standard evaluation metric for human emulation is move-matching accuracy, which is the rate at which, given a board position encountered in a real game, a model correctly predicts the move played by a human player. For ease of comparison with prior work in our main evaluation, we adopt the dataset of 884,049 positions curated by Zhang et al. (2025), which we call the ALLIE test set. This set was constructed by sampling Lichess blitz games from 2022 and removing the first 10 moves from each game, as these can be easily memorized, as well as removing positions that occur after the first time a player has fewer than 30 seconds on their clock, which tend to be noisier due to time pressure. During training, we retain the first 10 moves but discard moves made under time pressure in the same way. We describe the processing of training data in more detail in Appendix A.1.

### 4.2 TRAINING METHODOLOGY

We train a sequence of three increasingly large models, with 5M, 23M, and 79M parameters respectively, and refer to the largest of these as MAIA-3. At the 5M scale, we compare the performance of GAB to that of the absolute and relative biases described in Appendix C. We also train a smaller 3M-parameter Chessformer to gauge the extent to which GAB can reduce the need

Table 1: Main results for human modeling. The MAIA-3 family of models achieves new state-of-the-art performance with a fraction of the parameter count. MAIA$^\star$ denotes choosing the closest MAIA model to the active player's rating. GPT-3.5 is included for comparison following prior work.

| | Accuracy (%) | #Params | History | Search |
|---|---|---|---|---|
| MAIA-3-79M | **57.1 ± 0.1** | 79M | ✓ | ✗ |
| MAIA-3-23M | 56.6 ± 0.1 | 23M | ✓ | ✗ |
| MAIA-3-5M | 55.4 ± 0.1 | 5M | ✓ | ✗ |
| ALLIE-ADAPTIVE-SEARCH | 55.9 ± 0.1 | 355M | ✓ | ✓ |
| ALLIE-POLICY | 55.7 ± 0.1 | 355M | ✓ | ✗ |
| MAIA-2 | 52.0 ± 0.1 | 23M | ✗ | ✗ |
| MAIA$^\star$ | 51.6 ± 0.1 | 92M | ✗ | ✗ |
| GPT-3.5 | 53.7 ± 0.1 | 175B | ✓ | ✗ |

for model scale. For GAB models at the 5M and 3M scale, we replace the first linear projection and flattening layer of GAB with average pooling, as these are parameter-intensive at small scales. Initial experiments showed this to have only a minor effect on performance, decreasing move-matching accuracy by approximately $0.2\%$.

In chess, skill is modeled with the rating systems such as Elo, where Elo ratings vary roughly from 500 for weak players to 3000 for the strongest human players (exact numbers vary between different populations of players and rating system implementations). We condition human-emulating models on the skill levels of both players by prepending two "soft embeddings" of dimension 128, corresponding to the ratings of the players, to each of the 64 tokens. Following ALLIE, we compute an embedding $e_k$ for a rating $k$ as a linear interpolation between two learnable embeddings: a weak embedding ($e_{\text{weak}}$) corresponding to 0 and a strong engine-level embedding ($e_{\text{strong}}$) corresponding to 5000. Formally, we set $e_k = \gamma e_{\text{weak}} + (1 - \gamma)e_{\text{strong}}$, where $\gamma = \frac{5000-k}{5000}$. Representing the ratings as scalar inputs would achieve the same representational capacity, but this design enables the flexibility to, for example, model individual behavioral styles.

The final input for our human emulation models consists of 64 tokens, a concatenation of representations of the current and $n$ past board states and two strength embeddings of dimension 128. This results in a depth of $12 \times (1 + n) + 2 \times 128$, which is 352 for the $n = 7$ hyperparameter choice used in our main training and ablation runs. Despite the dimensions of this input being dominated by these embedding vectors, we did not find the choice of embedding dimension 128 to impact performance. Detailed information on our training setup can be found in Appendix A.1.

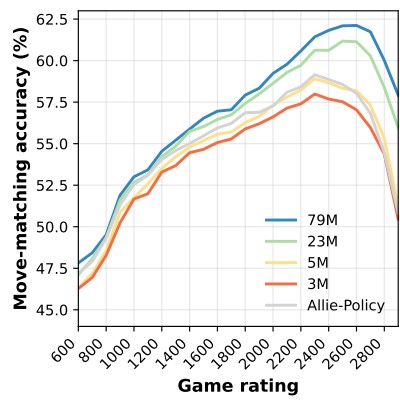

Figure 2: Move-matching accuracy on the ALLIE-AUGMENTED test set.

### 4.3 BOARD HISTORY

A full chess state is Markov when it includes side to move, castling rights, en passant availability, halfmove/repetition information, and the current board position. However, human play may depend on more than the current state: players form plans, reveal weaknesses, and make choices that may be predictable from previous actions. Prior human-emulation work varies in its use of history, with move-token methods like ALLIE conditioning on full game histories and square-based methods like MAIA and MAIA-2 omitting history entirely. We therefore condition models on the current and past $n$ board states. We find performance increases substantially from $n = 0$ to $n = 7$, with no significant gain from $n = 7$ to $n = 31$, so we use $n = 7$. Full ablations are in Appendix E.

### 4.4 RESULTS

Table 1 reports overall move-matching accuracies of MAIA-3 models on the ALLIE test set, demonstrating a significant improvement in move-matching performance and parameter efficiency [1]. Figure 2 shows that these gains hold across a wide range of skill levels, with the improvement increasing with rating. Our 79M-parameter and 23M-parameter models achieve move-matching accuracies of 57.1% and 56.6%, respectively, outperforming the state-of-the-art 355M-parameter searchless (55.7%) and search-enabled (55.9%) ALLIE methods at substantially smaller scales. Our 5M-parameter model reaches a move-matching accuracy of 55.4%, achieving results comparable to the previous state of the art with 70 times fewer parameters.

We plot the move-matching accuracies of MAIA-3 by the ratings of the active and opponent players in Figure 5. Results for ablating the position encoding are presented in Table 7, while results for ablating the number of history positions $n$ are reported in Table 8. The baseline MAIA-3-5M model significantly outperforms ablations equipped with absolute and relative biases, while MAIA-3-3M matches their performance at 30% fewer parameters and 40% fewer FLOPs. Chessformer enables a Pareto improvement across model scale, computation, and both policy and value metrics.

## 5 OPTIMIZING PLAYING STRENGTH: LEELA CHESS ZERO

We next apply Chessformer to raw chess playing strength in the searchless setting, where models select moves directly without tree search. We distill Leela Chess Zero into Chessformer models and ablate the position representation to isolate the contribution of Geometric Attention Bias (GAB). We then analyze the strongest resulting model, a 191M-parameter Chessformer that we call Leela-CF, and compare its searchless playing strength to prior searchless chess models. Finally, we show that these gains also translate to full engine play by integrating Leela-CF into full Leela configurations. We first run a controlled tournament comparing Leela Chess Zero configurations using either Leela-CF or the tournament-tested Leela-CNN convolutional network, and then report results from several elite computer chess tournaments in which Chessformer-equipped Leela Chess Zero configurations outperformed fields that included Stockfish, generally considered the strongest chess engine available.

### 5.1 TRAINING METHODOLOGY

Leela Chess Zero is an open-source recreation of AlphaZero, which iteratively teaches a randomly initialized neural network by generating self-play games between MCTS-augmented versions of that neural network. This neural network outputs a policy distribution that predicts the distribution of moves chosen by the MCTS algorithm and a value that predicts the outcome of the game. In effect, a model continually generates strong search-enabled oracles whose play is then distilled back into that model.

The most expensive component of the AlphaZero process by far is the generation of training games, which typically requires hundreds of model evaluations per position. We skip this step by fixing a dataset of self-play games from an April 2024 reinforcement learning run of Leela Chess Zero, keeping only those games that occurred once the model's strength had leveled off. That run used a 100M-parameter transformer model at 600 nodes per move with a square-based token representation and our GAB biases and attention policy. In this way, we move to the supervised setting, distilling a search-augmented version of one model into another. Initial experiments showed that Chessformer models trained on self-play games produced by convolutional neural networks and other Chessformer models reach virtually identical playing strengths, suggesting that the strength of the oracle is far more important than the model underlying its search process.

To motivate our design choices, we compare the performance of a 4M-parameter GAB model trained with this setup to those of the position encodings described in Appendix C at the same scale. We also train a Chessformer model with 2.5M parameters to demonstrate that GAB can outperform much larger models with simpler position encodings. Our oracle distillation setup is described in more detail in Appendix A.2.

---

[1] Results for MAIA* and GPT-3.5 are from Zhang et al. (2025).

## 5.2 EVALUATION METHODOLOGY

We consider two types of agents in our analysis: those that pick the highest-ranked move in the model's policy distribution, and those that evaluate each legal move and select the move that maximizes the resulting value prediction. The policy strategy requires a single model evaluation, while the value maximization strategy requires a model evaluation for each legal move. To estimate the floating point operations (FLOPs) per evaluation used by an agent of the value maximization type, we multiply the model FLOPs by 20, which is a conservative estimate of the average number of legal moves available in a position.

We construct agents from the 191M-parameter model using both strategies, denoting an agent by its model name followed by the strategy it uses (i.e., Leela-CF-policy and Leela-CF-value). We also construct agents from both strategies using a 195M-parameter convolutional neural network trained by the Leela Chess Zero project, which we call Leela-CNN. Our analysis includes additional models from Ruoss et al. (2024) that use the value maximization approach. We use the final checkpoints of their main runs and refer to them as AC-9M, AC-136M, and AC-270M, indicating their parameter counts.

Our evaluation setup is adapted almost exactly from Ruoss et al. (2024). We compare agents on both tournament strength and puzzle-solving ability. To measure the former, we play 200 games between each pair of agents and calculate relative Elo ratings using BayesElo (Coulom, 2008). We perform separate tournaments for the main and ablation runs, anchoring the Elo value of the absolute position encoding to 0 and the Elo value of AC-270M to the value reported by Ruoss et al. To measure puzzle-solving ability, we report the accuracy of these models on a test set of 10,000 puzzles curated by that work. For our ablation runs, we also report accuracy and loss metrics on the value and policy heads. These were calculated on a set of 1.4M test positions.

## 5.3 RESULTS

As shown in Table 2, Leela-CF-policy has the lowest computational cost while outperforming all non-Chessformer baselines in Elo and remaining competitive on puzzles; Leela-CF-value is strongest overall. We also see performance on the puzzle set approaching saturation, suggesting that new metrics for evaluation might be needed.

Our ablation results, reported in Table 5, show a monotonic improvement in performance from absolute to relative bias to GAB encodings at comparable scale. GAB outperforms the baseline absolute position encoding by 1.9% on policy accuracy, 0.3% on game outcome prediction accuracy, 3.2% on puzzle-solving accuracy, and 83 Elo rating points in tournament strength. We note that the maximum possible policy and value accuracies are well under 100%, both because Leela's self-play methodology is inherently nondeterministic and because there are positions with a variety of best moves. GAB enables a model to perform on par with the absolute position ablation at around 60% the parameter count and computation.

Table 2: Results for raw playing strength. Chessformer achieves significant gains in both Elo and puzzle-solving accuracy, using fewer FLOPs than competing models.

|  | Elo | Puzzles (%) | FLOPs |
|---|---|---|---|
| Leela-CF-policy (ours) | $2374 \pm 37$ | $93.5 \pm 0.5$ | 7.6B |
| Leela-CF-value (ours) | $2466 \pm 36$ | $97.2 \pm 0.3$ | 152B |
| AC-9M | $2044 \pm 42$ | $86.2 \pm 0.7$ | 14.2B |
| AC-136M | $2257 \pm 36$ | $92.7 \pm 0.5$ | 215B |
| AC-270M | $2299 \pm 36$ | $94.2 \pm 0.5$ | 427B |
| Leela-CNN-policy | $2096 \pm 40$ | $82.1 \pm 0.8$ | 12.5B |
| Leela-CNN-value | $2168 \pm 36$ | $92.5 \pm 0.5$ | 249B |

## 5.4 ENGINE STRENGTH

To demonstrate that our Chessformer models have the capacity to push general engine strength, we run a tournament between configurations of the Leela engine paired with either the Leela-CNN model, a 195M-parameter convolution-based model previously used by Leela in tournaments, or Leela-CF, a 191M-parameter Chessformer. We played 2000 games between these configurations at three time controls, described in further detail in Appendix B. As shown in Table 6, the Leela-CF Chessformer model consistently increases the playing strength of Leela Chess Zero by over 100 Elo. To contextualize these gains, the difference in playing strength between Stockfish versions 16 and 17, corresponding to 14 months of continuous development progress, was measured at around

46 Elo under a similar testing setup (Stockfish Team, 2024). This is especially notable due to the difficulty of improving top engines; the Stockfish project continually employs thousands of CPU cores to test over 10,000 potential improvements per year (Stockfish Team, 2022).

Configurations of Leela Chess Zero equipped with Chessformer models defeated Stockfish, a perennial champion, in several online computer chess tournaments hosted by the Top Chess Engine Championship (TCEC). These victories included the single-elimination TCEC Cup 11 tournament, where 32 engines faced off in brackets, and two Swiss-system tournaments, the TCEC Swiss 6 and 7 events, each of which had around 40 contestants. In the cup event, Leela advanced through every round and won by defeating Stockfish in the final. Leela also won both Swiss events.

## 6 INTERPRETABILITY

Chess has emerged as a useful testbed for mechanistic interpretability, which seeks to uncover the internal mechanisms by which AI models operate. Chessformer's domain-grounded architecture supports this goal by making both attention patterns and intermediate activations naturally attributable to specific board squares. As a preliminary investigation of these interpretability benefits, we analyze how GAB and dot-product attention interact to shape the model's representations and move predictions.

### 6.1 GEOMETRIC AND SEMANTIC STRUCTURE IN ATTENTION

How do GAB and dot-product attention divide representational work in Chessformer? Since the final attention logits are the sum of learned dot-product attention and position-dependent GAB biases, each component may capture different kinds of structure. In particular, we ask whether GAB biases primarily encode square-specific geometric information, while dot-product attention captures more global semantic information about the position.

To test this, we measure how much the rows of the GAB and dot-product attention maps vary both across positions and within individual positions. We compute MAIA-3's attention maps on 30,000 randomly sampled positions from Lichess blitz games played in June 2019. For each component, we calculate *between-position consistency* as the average correlation between attention-map rows from distinct positions, holding the query square and head fixed, and averaging across query squares, position pairs, and heads. We calculate *within-position consistency* as the average correlation between attention-map rows for distinct query squares within the same position, averaging across query-square pairs, positions, and heads. In both cases, each component is compared only with itself; we do not directly correlate GAB and dot-product attention maps.

Table 3 shows that GAB biases are relatively consistent across positions, but highly variable across query squares within a position. In contrast, dot-product attention varies substantially across positions, but is much more consistent across query squares within a position. This suggests that GAB primarily captures square-specific geometric structure, while dot-product attention reflects more global semantic information about the board state.

Although GAB is more stable across positions than dot-product attention, it is not fixed: its biases still adapt meaningfully to the position context. Figure 3 illustrates this behavior in one head, where the GAB component transitions from modeling a broad range of piece movement in the early game to emphasizing king movement in the late game. We provide additional examples of GAB and dot-product attention maps in Appendix F.2.

### 6.2 IDENTIFYING INTERPRETABLE FEATURES

A prominent approach to transformer interpretability trains sparse autoencoders (Bricken et al., 2023) or transcoders (Dunefsky et al., 2024) on MLP activations, then interprets the learned features using the inputs that maximally activate them. We apply this approach in chess, building on prior mechanistic interpretability work in the domain (Karvonen, 2024; Mei, 2025; Jenner et al., 2024).

Chessformer's square-token architecture enables a more fine-grained version of this analysis. Rather than interpreting a feature only from the positions that activate it most strongly, we can also attribute each activation to a specific board square. To explore this property, we train a cross-layer

transcoder on MAIA-3; training details are given in Appendix A.4. The resulting features are highly interpretable and span a rich set of chess concepts. Many correspond to tactical motifs such as forks and pins, with their strongest activations localized to the squares involved in those tactics. Others are difficult to interpret from their top-activating positions alone, but become clear once activations are localized to individual squares.

Because space precludes visualizing all 8,192 transcoder features, we annotate the top-activating positions for the first 20 features from each transcoder layer in Figures 8 through 11. Since feature order is arbitrary, these provide an approximately unbiased sample. The prevalence of interpretable features, together with the usefulness of square-level activation attribution, suggests that Chessformer's architecture makes mechanistic analysis of chess models substantially more natural.

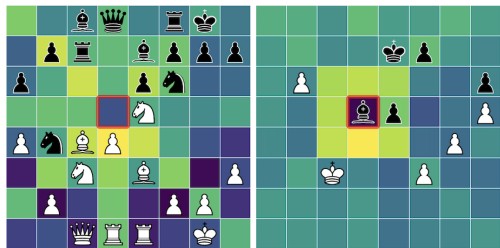

Figure 3: GAB bias maps in L14H11 of Leela-CF in the early and late game. The GAB bias for this head transitions from modeling a wide range of movement in the early game (left) to king movement in the late game (right).

Table 3: Average correlations of attention rows within and between positions for Geometric Attention Bias (GAB) and dot-product attention (DPA). GAB exhibits higher variability within positions than DPA, but lower variability between positions for a given query square.

| Mean correlation | GAB | DPA |
|---|---|---|
| Between positions | 0.770 | 0.230 |
| Within positions | 0.005 | 0.816 |

## 7 DISCUSSION

We introduce Chessformer, a domain-aligned transformer architecture for chess that improves both raw playing strength and human move prediction, and does so in a naturally interpretable way. Across two major settings, Chessformer advances the Pareto frontier of chess modeling: MAIA-3 achieves state-of-the-art human move prediction with fewer than one-fourth the parameters of prior methods, and Leela-CF improves searchless playing strength enough to translate into stronger full-engine play when integrated into Leela Chess Zero. MAIA-3 reaches 57.1% move-matching accuracy, surpassing both searchless and search-enabled ALLIE baselines. For playing strength, our ablations show consistent gains from increasingly domain-aligned position representations: GAB improves policy accuracy, value accuracy, puzzle-solving accuracy, and Elo over absolute encodings, while matching that baseline with around 60% of the compute.

The broader lesson is that human compatibility and task performance need not be opposing goals. In chess, aligning the model architecture with the structure of the domain improves both mastery and behavioral modeling. This same alignment also makes mechanistic analysis more natural, since attention patterns and activations can be attributed directly to board squares. Chessformer therefore suggests a general recipe for structured decision domains: models may become stronger, more human-compatible, and more interpretable when their internal representations better reflect the geometry of the task.

Several limitations point to concrete next steps. GAB is currently specialized to chess, and its benefits may depend on domains where geometric relations are central. Extending this approach to other structured decision problems will require identifying the right domain symmetries, tokenizations, and action representations. Our interpretability results are also preliminary. Deeper mechanistic analysis is needed to understand how the complementary roles of GAB and dot-product attention support planning, evaluation, and human-like play.

### ACKNOWLEDGMENTS

We gratefully acknowledge support from the Natural Sciences and Engineering Research Council of Canada (NSERC), the Canada Foundation for Innovation (CFI), and the Ontario Research Fund (ORF). We also thank the Leela Chess Zero team for technical assistance and access to hardware.

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

## A    IMPLEMENTATION DETAILS

### A.1    HUMAN EMULATION

All human-prediction models were trained with the AdamW optimizer on the dataset described in Section 4. During training, we sample 32 positions per game at random, or take all of them if 32 positions are not available. All runs have 8 layers, head dimension 32, and an MLP expansion factor of 2. Our 3M, 5M, 23M, and 79M models have embedding dimensions of 192, 256, 512, and 1024, respectively. GAB is configured with $d_1 = 32$ and $d_2 = d_3 = 128$ for the 23M and 79M models and average pooling and $d_2 = d_3 = 64$ for the 5M model and all its ablations. All runs were trained for 1 million steps with a cyclic cosine annealed learning rate schedule. The 23M and 79M main runs were performed on 8 A100 GPUs, taking a few days and a week respectively, and all other runs were trained on 2 A100 GPUs for around a week.

Because low-rated games are overrepresented in our training dataset, we downsample this dataset during training to ensure that skill levels are equally represented. In particular, the rating spectrum is divided into 22 bins: 20 bins uniformly spanning 600 to 2600 Elo with 100-point intervals, plus two bins for players rated below 600 or above 2600. For each game, we compute the average Elo of the two players and assign the game to the corresponding bin. We organize the raw game data into chunks of 20,000 games. For each chunk, we iterate through games sequentially and distribute them into bins until each bin accumulates 10 games. The process terminates when either all games in the chunk are consumed or all bins reach 10 games. This encourages equal representation across skill levels, removing the bias toward low-rated games.

The ALLIE test set is suitable to measure overall performance, and all overall performance metrics we report on the human emulation task employ it. However, it lacks positions at very high and low skill levels and therefore cannot be used to form statistically significant conclusions at these skill levels. For example, it has only 205 positions rated above 2850, which would result in a margin of error of over 7%. To rectify this, we augment the ALLIE test set with very highly and lowly rated Lichess blitz games from August and September 2025 to form the ALLIE-AUGMENTED dataset, which we use to compare modeling performance across skill levels— it is used in Figure 5, Figure 6, and Figure 7, and nowhere else.

To obtain the ALLIE-AUGMENTED test set from the ALLIE test set, we first take rating intervals of length 100 between 550 and 2950 in which the ALLIE test set contains fewer than 20 thousand positions. These intervals are chosen so that they correspond to the bins in Figure 6 and Figure 7, which round the game rating. We initialize empty buckets for each such interval and iterate through September and August 2025 Lichess blitz games, adding the game to the corresponding bucket so long as it has less than a thousand games. These games are added on top of the ALLIE test set to form the final ALLIE-AUGMENTED set of 1,087,778 positions.

### A.2    PLAYING STRENGTH

As described in Section 5, we train in the supervised setting on games generated from a past reinforcement learning run of the Leela Chess Zero project. This strategy works well in practice and was used to train Leela-CF, which surpassed the model that generated its training data, as well as each of the models used in the victories against Stockfish described in Section 5.

To maximize compatibility with the Leela Chess Zero infrastructure, we follow their input scheme as closely as possible. We form 64 tokens of depth 96 for the current and past 7 board states, following Section 3.1. We also concatenate indicators of whether each of the current and past 7 board states was a repetition, whether each of the 4 castling options are available, whether black is to move, and the number of plies since the last capture or pawn move, divided by 100. Finally, we concatenate a 0 and 1, which are relics that were originally intended to allow convolutional models to detect edges. This information is concatenated to each token, giving 64 pre-embedding tokens of depth $(96 + 8 + 4 + 1 + 1 + 2) = 112$. Initial experiments did not show these additional inputs to alter performance.

All Chessformer models trained for this task used the Nadam optimizer with $\beta_1 = 0.9$, $\beta_2 = 0.98$, $\epsilon = 10^{-7}$, and gradient clipping 10. Following the Leela Chess Zero setup, checkpoints were

Table 4: Human Move-Matching Training Configuration for Reproducibility

| Parameter | Value |
|---|---|
| *Training Setup* | |
| batch_size_train | 128 |
| batch_size_val | 16 |
| gradient_accumulation_steps | 4 |
| num_workers | 8 |
| *Optimization* | |
| lr | $5 \times 10^{-5}$ |
| min_lr | $1 \times 10^{-5}$ |
| wd (weight decay) | $1 \times 10^{-6}$ |
| grad_clip_norm | 3.5 |
| warmup_steps | 1,000 |
| cosine_cycles | 50,000 |
| refresh_lr_scheduler | true |
| *Mixed Precision* | |
| use_amp | true |
| amp_init_scale | 256 |
| amp_max_scale | 8,192 |
| amp_growth_factor | 1.5 |
| amp_growth_interval | 2,000 |
| amp_backoff_factor | 0.5 |
| *Loss Weights* | |
| value_coefficient | 0.1 |

calculated using Stochastic Weight Averaging (Izmailov et al., 2018), which incrementally boosted performance.

The 191M-parameter Leela-CF model has hidden dimension 1024, MLP dimension 1536, a head size of 32, and 15 layers, and GAB is configured $d_2 = d_3 = 256$, and the first linear projection and flattening layers are replaced with average pooling, which initial experiments showed greatly improves parameter efficiency at small scales at the cost of a slight performance degradation. It was trained for 6 million steps, with the learning rate initialized to $2 \times 10^{-3}$ and dropped to $3 \times 10^{-4}$ at 4.49 million steps and $3 \times 10^{-5}$ at 5.47 million steps. The 195M-parameter Leela-CNN is a squeeze-excitation ResNet (Hu et al., 2018; He et al., 2016) with 512 filters and depth 40.

Table 5: Ablation results for raw playing strength. Accuracies and Elo values are reported with 95% confidence intervals

| | Loss | | Accuracy (%) | | Puzzles (%) | Elo | FLOPs | #Params |
|---|---|---|---|---|---|---|---|---|
| | Policy | Value | Policy | Value | | | | |
| Absolute | 0.363 | 0.567 | $56.6 \pm 0.1$ | $88.7 \pm 0.1$ | $61.0 \pm 1.0$ | $0 \pm 18$ | 210M | 3.67M |
| Relative bias | 0.346 | 0.565 | $57.5 \pm 0.1$ | $88.8 \pm 0.1$ | $63.2 \pm 1.0$ | $40 \pm 18$ | 210M | 3.67M |
| GAB-2.5M | 0.360 | 0.567 | $57.0 \pm 0.1$ | $88.7 \pm 0.1$ | $61.5 \pm 1.0$ | $-4 \pm 18$ | 131M | 2.51M |
| GAB-4M | 0.330 | 0.562 | $58.5 \pm 0.1$ | $89.0 \pm 0.1$ | $64.2 \pm 1.0$ | $83 \pm 18$ | 228M | 4.01M |

We train ablations with the three position representations described in Section 3 with 8 layers, embedding dimension 256, head dimension 32, and MLP dimension 256. GAB is configured with $d_1 = 8$ and $d_2 = d_3 = 32$. The 2.5M model has embedding dimension and MLP dimension 192, with all else held constant. Each was trained for 1.4 million steps on a single A100 GPU with a batch size of 2048 in approximately four days. The learning rate was held constant at $5 \times 10^{-4}$.

## A.3 SPECIAL MOVES

A source and destination square are sufficient to represent all moves that can occur within the rules of chess, with some exceptions. When a pawn advances to the last rank of the board, it must be promoted to a knight, bishop, rook, or queen. To represent these special moves, we apply a linear projection to the key vectors for squares in the last rank, generating an additive bias for each possible promotion piece. This bias is then applied to the logits representing all possible traversals between the penultimate rank and the promotion rank to generate additional logits for each possible promotion. Following the standard in computer chess, en passant captures are encoded as diagonal moves, and castling is encoded as the king moving two spaces horizontally.

```python
def sm_bias(x: torch.Tensor) -> torch.Tensor:
    B = x.shape[0]
    y = sm1(x)  # (B, 64, d_1)
    y = y.reshape(B, -1)  # (B, 64d_1)
    y = sm_act(sm2(y))  # (B, d_2)
    y = ln1(y)
    y = sm_act(sm3(y))  # (B, H*d_3)
    y = ln2(y).view(B, num_heads, gen_size)  # (B, H, gen_size)
    b = torch.einsum("bhi,oi->bho", y, self.posenc_weight)
    return b.view(B, self.num_heads, 64, 64)
```

Figure 4: Torch-like pseudocode for GAB.

## A.4 TRANSCODER TRAINING

For interpretability purposes, we train a cross-layer transcoder on MLP activations collected from layers 3 and 4 (in other words, the 4th and 5th layers) of an earlier checkpoint of MAIA-3. The transcoder consists of encoders for each layer and decoders going between the two layers (including between each layer and itself), trained on reconstruction and sparsity loss. We train only on these layers because of compute constraints and preliminary investigations that showed that these layers tended to contain the most interpretable representations. It is common practice to train sparse autoencoders and transcoders on medium-depth layers of models, as these layers often contain the most interpretable representations (Gurnee et al., 2023).

Let the base transformer have $L_{\text{base}}$ layers, and let $S = \{\ell_0 < \ell_1 < \cdots < \ell_{K-1}\}$ be a subset of $K$ layers on which we train the transcoder (for example, $S = \{3, 4\}$ when training only between layers 3 and 4). For each $\ell_k \in S$ we denote by $\mathbf{R}_{\ell_k}^{\text{pre}} \in \mathbb{R}^{B \times T \times D}$ the pre-MLP residual stream and by $\mathbf{M}_{\ell_k} \in \mathbb{R}^{B \times T \times D}$ the corresponding MLP output. After per-layer standardization over batch and tokens, we write

$$\mathbf{X}_k \in \mathbb{R}^{B \times T \times D}, \quad \mathbf{Y}_k \in \mathbb{R}^{B \times T \times D}, \quad k = 0, \ldots, K-1,$$

for the normalized inputs and targets. The transcoder operates on the index set $\{0, \ldots, K-1\}$, with indices $i, j$ referring to layers $\ell_i, \ell_j \in S$.

$$\text{Encoder:} \quad \mathbf{Z}_i = \mathbf{X}_i \mathbf{W}_i^{(e)} + \mathbf{b}_i^{(e)} \in \mathbb{R}^{B \times T \times M}, \tag{1}$$

$$\mathbf{A}_i = \text{ReLU}(\mathbf{Z}_i - \boldsymbol{\tau}_i) \in \mathbb{R}^{B \times T \times M}, \tag{2}$$

where $\mathbf{W}_i^{(e)} \in \mathbb{R}^{D \times M}$, $\mathbf{b}_i^{(e)} \in \mathbb{R}^M$, and $\boldsymbol{\tau}_i \in \mathbb{R}^M$ is a learned threshold broadcast over batch and tokens.

$$\text{Decoders (for all } 0 \leq i \leq j < K): \quad \widehat{\mathbf{Y}}_{i \to j} = \mathbf{A}_i \mathbf{W}_{i \to j}^{(d)} + \mathbf{b}_{i \to j}^{(d)} \in \mathbb{R}^{B \times T \times D}, \tag{3}$$

$$\widehat{\mathbf{Y}}_j = \sum_{i=0}^{j} \widehat{\mathbf{Y}}_{i \to j}, \tag{4}$$

where $\mathbf{W}_{i \to j}^{(d)} \in \mathbb{R}^{M \times D}$ and $\mathbf{b}_{i \to j}^{(d)} \in \mathbb{R}^D$.

$$\text{Reconstruction loss:} \quad \ell_j^{\text{MSE}} = \frac{1}{BTD} \left\| \widehat{\mathbf{Y}}_j - \mathbf{Y}_j \right\|_2^2, \tag{5}$$

$$\mathcal{L}_{\text{recon}} = \frac{1}{K} \sum_{j=0}^{K-1} \ell_j^{\text{MSE}}. \tag{6}$$

$$\text{Decoder-weighted sparsity penalty:} \quad \pi_{i,f} = \frac{1}{K-i} \sum_{j=i}^{K-1} \left\| \mathbf{W}_{i \to j}^{(d)}[:,f] \right\|_2, \quad f = 1, \dots, M, \tag{7}$$

$$s_i = \frac{1}{BTM} \sum_{b=1}^{B} \sum_{t=1}^{T} \sum_{f=1}^{M} \tanh\!\left( c\, \pi_{i,f}\, (\mathbf{A}_i)_{b,t,f} \right), \tag{8}$$

$$\mathcal{L}_{\text{sparse}} = \lambda \cdot \left( \frac{1}{K} \sum_{i=0}^{K-1} s_i \right), \tag{9}$$

where $\lambda > 0$ and $c > 0$ are scalar hyperparameters.

$$\text{Total loss:} \quad \mathcal{L}_{\text{total}} = \mathcal{L}_{\text{recon}} + \mathcal{L}_{\text{sparse}}. \tag{10}$$

We use a batch size of 22 games, a learning rate of $5 \times 10^{-5}$, and an expansion factor of 8. For training data, we use blitz games from lichess played during July 2019, filtered in the exact same way as in our base model training pipeline. We use the same data to sample the top-activating tokens for each feature. At the end of training, our transcoder achieves a reconstruction MSE of 1.6% and a sparsity of 0.90.

## B  IMPLEMENTATION DETAILS FOR TOURNAMENT

Table 6: Tournament performance of Leela-CF Chessformer against Leela-CNN convolution model. The time control is chosen so that $N$ is roughly the number of playouts the Leela-CNN model performs during each game. Elo values are reported with 95% confidence intervals.

| $N$ | Elo Gain | Wins | Losses | Draws |
|------|----------|------|--------|-------|
| 160k | $112 \pm 7$ | 846 | 223 | 931 |
| 320k | $111 \pm 7$ | 806 | 186 | 1008 |
| 640k | $105 \pm 7$ | 779 | 190 | 1031 |

To gauge the impact of Chessformer on raw engine strength, we perform a tournament between versions of the Leela Chess Zero engine configured with either the Leela-CNN model, a 195M-parameter convolution-based model previously used by Leela at tournaments, or Leela-CF, a 191M-parameter Chessformer. Because of the high computational load of long engine analyses, we use the distributed testing framework OpenBench (Grant) to run 2000 games between these configurations, so that each of 8 RTX 4090 and 4 A100 GPUs runs a single game at a time with engines alternating use of the hardware. Games are played in pairs starting from positions sampled from *UHO_Lichess_4852_v1.epd*, a book of 2.6 million unbalanced human openings curated by the Stockfish project.

To calculate the base time control $T$ for each GPU, we benchmark the speed of Leela-CNN and set $T$ to an estimate of the amount of time the GPU would take to perform $N$ MCTS playouts, varying the value of $N$ to determine the effect of thinking time on the performance difference. The increment is set to $T/100$, and Leela Chess Zero is allowed to use its time according to its time management algorithm. In other words, $N$ is an estimate of the total number of playouts the Leela-CNN configuration performs over the course of the game. On average, the speed of Leela-CNN on these GPUs is approximately 20,000 playouts per second, so $T \approx N/20000$ seconds. Calculating the time control dynamically for each worker to adjust for variations in processing power is standard in modern distributed engine testing frameworks like Stockfish's Fishtest (Stockfish Team).

## C  POSITIONAL ENCODING BASELINES

**Absolute Position Embeddings**    Perhaps the simplest choice of position encoding is the absolute position embedding, which consists of adding a learned embedding to each token and was notably used by GPT-2 (Radford et al., 2019). Formally, given a sequence of token embeddings $\mathbf{x}_1, \ldots \mathbf{x}_n$, one applies

$$\mathbf{x}_i \mapsto \mathbf{x}_i + \mathbf{c}_i \tag{11}$$

prior to the transformer sublayers.

**Relative Position Biases**    Unlike absolute position embeddings, relative position encodings model positional information based on the relative displacement between tokens. One simple variant introduces relative biases $f_k$ which are added to the attention logits:

$$e_{ij} = \frac{(\mathbf{x}_i W^Q)(\mathbf{x}_j W^K)^T}{\sqrt{d}} + f_{i-j} \tag{12}$$

We consider the two-dimensional analog of this technique, where a square on the chessboard is assigned coordinates $(i, j)$, with $i$ and $j$ ranging from 0 to 7. The bias for querying square $(i_1, j_1)$ and key square $(i_2, j_2)$ is thus $f_{(i_2-i_1, j_2-j_1)}$, where $f_{a,b}$ is defined for $-7 \leq a, b \leq 7$. This adds $15 \times 15$ parameters per attention head.

## D  TOKENIZATION

A number of tokenization schemes have been proposed for chess. We review some of these and attempt to give insight into why our recipe, a square-based representation with a strong position encoding, significantly outperforms them.

The MAIA-2 architecture consists of a series of convolution blocks operating on a square-based visual board representation, followed by self-attention layers that process the depth-64 output channels as tokens. Though an interesting design choice, this does not align with the standard, well-tested recipe of transformers in domains like vision and language, which tokenizes inputs by partitioning them in space rather than through internal model representations.

Move-token formulations like ALLIE, on the other hand, rely on the established methodology of language modeling but lack another vital property: *specialization*. Processing inputs in parallel should allow a model to "divide and conquer", so that the overall computation is split into units that are processed with the same parameters. However, there is emerging evidence that move-token models do not specialize effectively and instead simply reconstruct the board state at each token (Mei, 2025; Karvonen, 2024). It is also not intuitively clear why a trajectory-based representation should be natural in the Markovian domain of chess.

## E  ADDITIONAL ANALYSIS FOR HUMAN EMULATION

Here we provide additional analysis on our human emulation results. We first ablate $n$, the number of past positions concatenated to the current one to form the input, at the 5M scale. Interestingly, as shown in Table 8, there is a large increase in performance between $n = 0$ and $n = 7$, but no significant difference between $n = 7$ and $n = 31$. The improvement is concentrated at low game ratings and decreases steadily up to high ratings. This contrasts with the effect of model size on move-matching performance, shown in Figure 6 and Figure 7. The impact of scale on modeling performance is several times higher for strong play than it is for weak play.

To ensure that positions in which no or little history information is available remain in-distribution, we mask out during training a uniformly random amount of history information with probability 5%. In initial experiments, this was found to negligibly affect performance ($< 0.05\%$ reduction in move-matching accuracy) relative to always retaining all $n$ past board states.

Despite marked improvements in parameter and compute efficiency, MAIA-3-79M improves on the state-of-the-art move-matching accuracy by only 1.2%. We postulate that despite Chessformer's

Table 7: Position encoding ablations for human emulation.

| | Loss | | Accuracy (%) | | FLOPs | #Params |
|---|---|---|---|---|---|---|
| | **Policy** | **Value** | **Policy** | **Value** | | |
| Absolute | 1.418 | 0.754 | $54.7 \pm 0.1$ | $62.6 \pm 0.1$ | 268M | 4.58M |
| Relative bias | 1.420 | 0.754 | $54.6 \pm 0.1$ | $62.6 \pm 0.1$ | 268M | 4.58M |
| MAIA-3-3M | 1.419 | 0.739 | $54.8 \pm 0.1$ | $62.6 \pm 0.1$ | 164M | 2.98M |
| MAIA-3-5M | 1.387 | 0.736 | $55.4 \pm 0.1$ | $62.8 \pm 0.1$ | 276M | 4.91M |

modeling capability, it runs into a performance ceiling at low and intermediate skill levels. Play at these skill levels is unsophisticated and easy to model, but inconsistent and stochastic enough that the accuracy appears to saturate at around 50%. MAIA-3-79M shines, however, at modeling highly skilled play, advancing the searchless state of the art by up to 5% for very strong play. Prior work has struggled to emulate strong play, often relying on search to shore up weak human-aligned models. That MAIA-3 not only outperforms even search-enabled methods but achieves its largest gains at these very high skill levels suggests that our methodology jointly enables both human alignment and mastery.

Table 8: Move-matching accuracy by history length for human emulation. The value reported is $n$, the number of past positions excluding the current one inputted into the model.

| History | Accuracy (%) |
|---|---|
| 0 | $54.0 \pm 0.1$ |
| 7 | $\mathbf{55.4 \pm 0.1}$ |
| 31 | $\mathbf{55.4 \pm 0.1}$ |

To understand why this is the case, we decompose the error for human emulation into aleatoric uncertainty, the amount of uncertainty that is inherent to the task, and epistemic uncertainty, the amount of uncertainty that can be reduced through stronger models. Low-rated play tends to be unsophisticated, reducing the amount of improvement available from better modeling approaches, but stochastic and inconsistent, reducing the ceiling on predictability. In other words, it has high aleatoric uncertainty, explaining the low accuracies for these players but low epistemic uncertainty, explaining the minor gains provided by scale. In contrast, highly skilled play is more accurate and thus more consistent, but more difficult to predict due to its sophistication; it has low aleatoric uncertainty but high epistemic uncertainty, giving plenty of room for improvement. History information likely improves move-matching performance by providing information about the player and reducing the aleatoric uncertainty of the task, particularly for low-rated play where it is higher.

With this in mind, we believe that future human emulation work should focus on highly skilled play, where the performance ceiling appears to be much higher. One interesting question is whether the maximum possible accuracy on this task increases monotonically with the game rating, aligning with our intuition about the consistency of strong players, or drops off for ratings above 2600, in line with our observed performance.

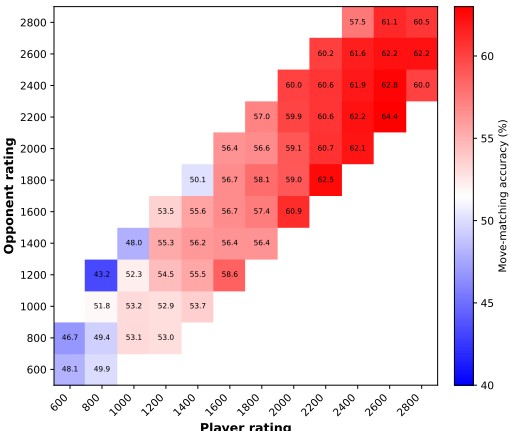

Figure 5: Move-matching accuracies of MAIA-3 for pairs of skill levels on the ALLIE-AUGMENTED test set, described in Appendix A.1.

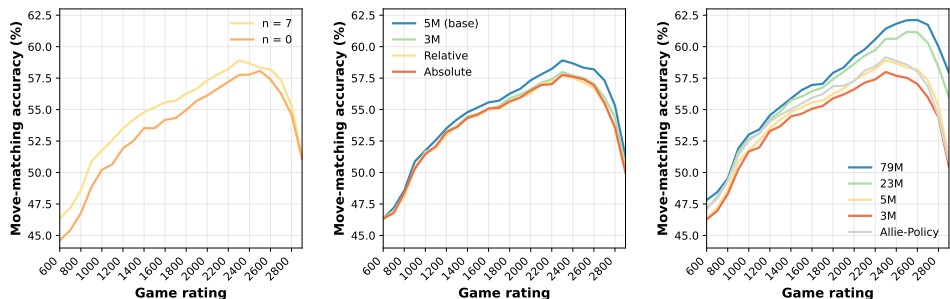

Figure 6: Human move-matching accuracies on the ALLIE-AUGMENTED test set by number of history positions $n$ (left), position encoding (middle), and scale (right). History information helps most for weaker play, while scale and effective position encodings have a large effect for stronger play. We omit results for $n = 31$ history positions as they are virtually identical to those for $n = 7$, and also omit ALLIE-ADAPTIVE-SEARCH due to compute constraints. For all game ratings other than 2900, the margin of error is less than a percent.

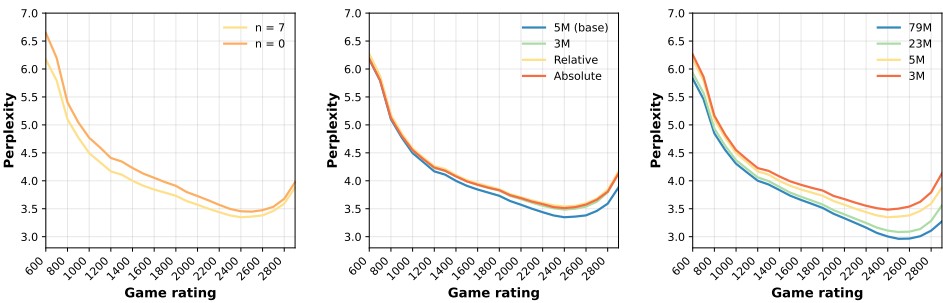

Figure 7: Human move-matching perplexity on the ALLIE-AUGMENTED test set by number of history positions $n$ (left), position encoding (middle), and scale (right). We omit results for $n = 31$ history positions as they are virtually identical to those for $n = 7$.

# F ADDITIONAL RESULTS

## F.1 TOP-ACTIVATED TOKENS FOR TRANSCODER

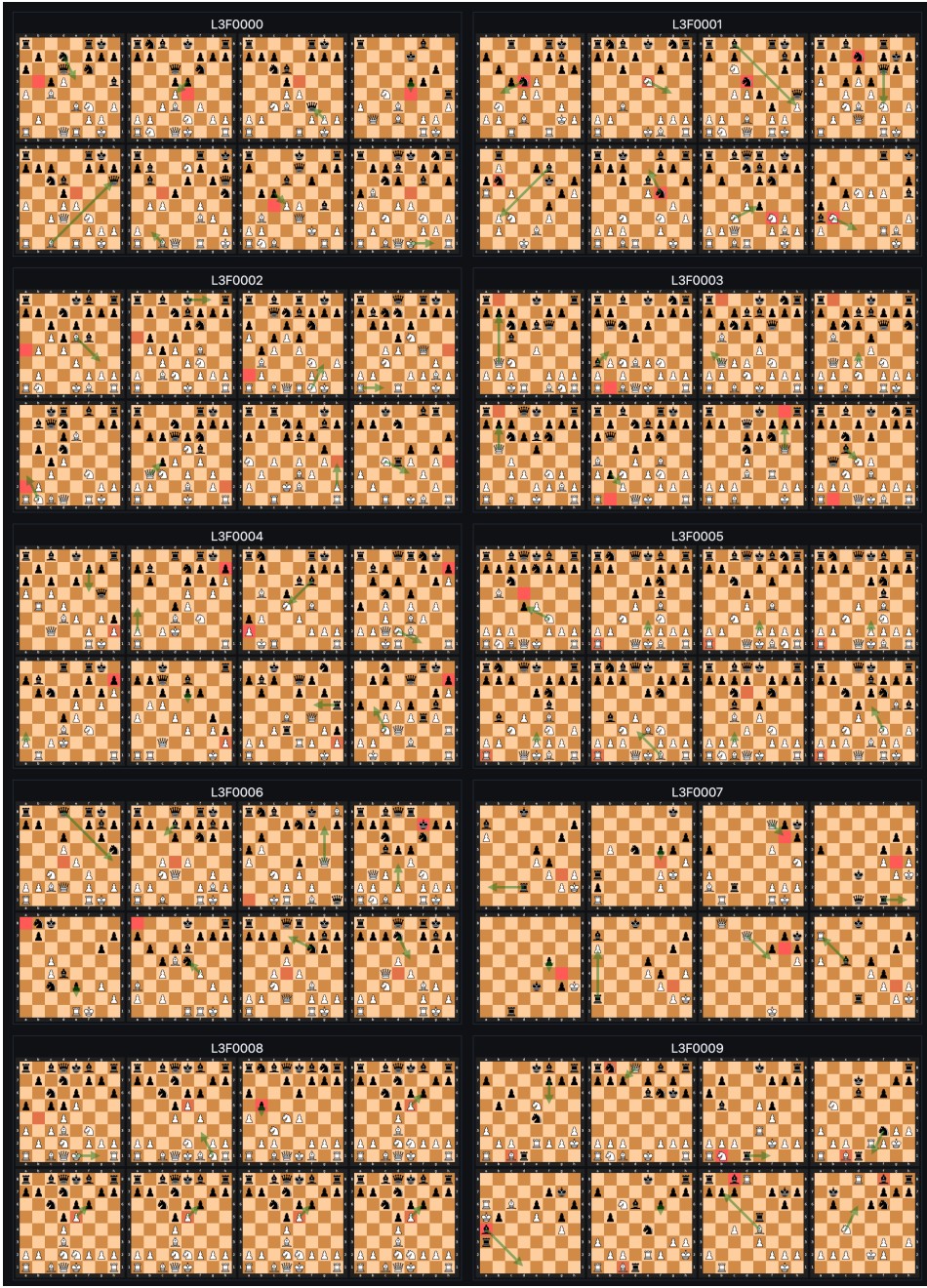

Figure 8: **Annotations for features 0-9 of layer 3.** L3F0000: Square that the active player can advance a pawn to in order to attack an enemy bishop. L3F0001: Active player's knight, usually under attack. L3F0002: Square on the side of the board that is controlled by the active player's rook or queen. L3F0003: Vacant square adjacent to a rook in the corner. L3F0004: An enemy pawn in the corner, in front of the active player's pawn, sheltering the opponent's king. L3F0005: Either a rook in the corner or a center pawn movement option. L3F0006: Not interpretable. L3F0007: A square diagonally adjacent to the opponent's king that is controlled by the active player's pawn. L3F0008: A square contested by both friendly and enemy pawns. L3F0009: Enemy minor piece on the edge of the board pinned to an enemy rook.

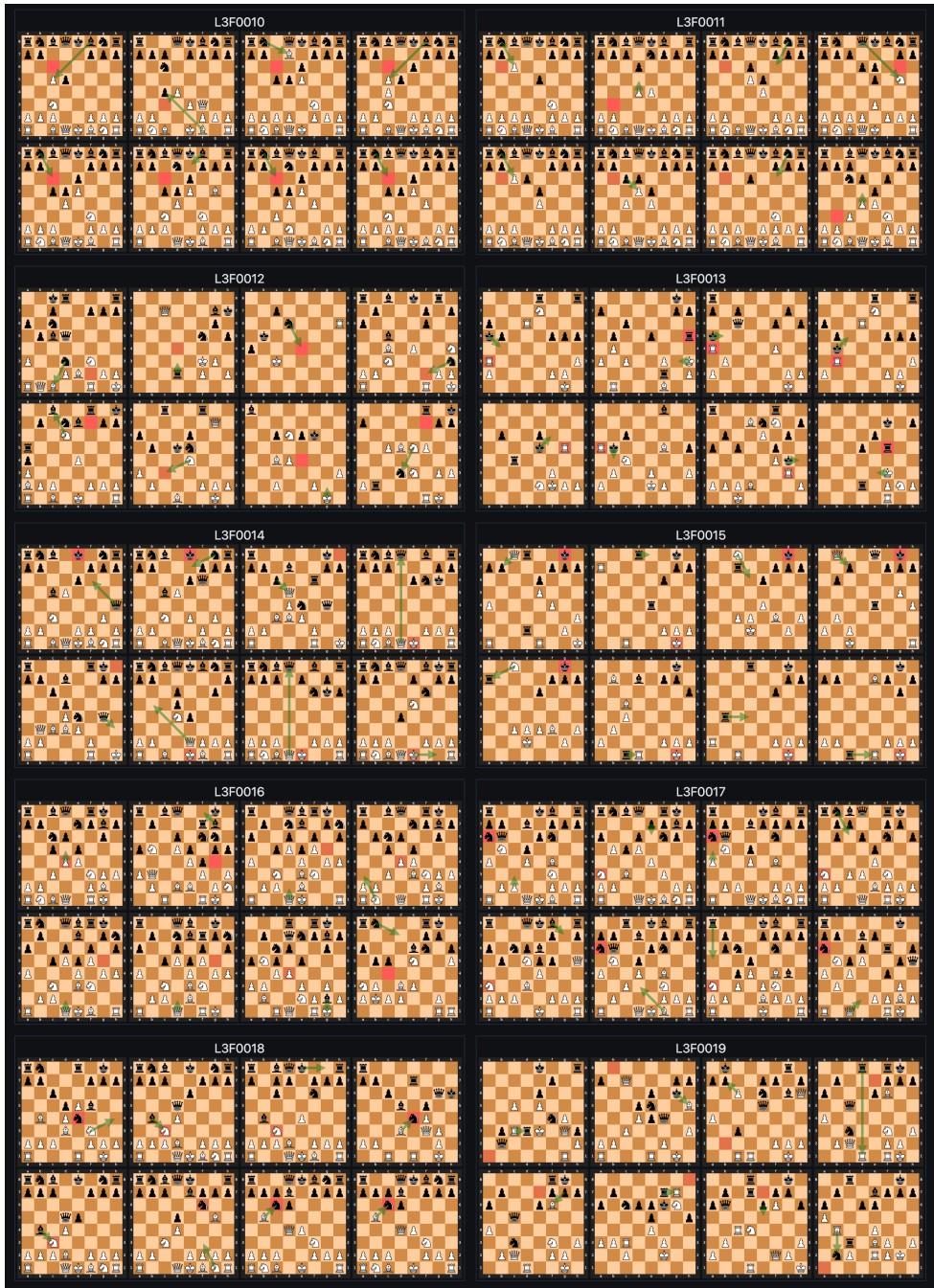

Figure 9: **Annotations for features 10-19 of layer 3 of** MAIA-3. L3F0010: Queenside activation square for active player's knight in Queen's gambit structures. L3F0011: Square on b3, b6, f3, or f6 in the opening that have been weakened by the lack of a supporting pawn. L3F0012: Square that the active player's knight can move to to give check. L3F0013: Enemy rook checking the active player's king. L3F0014: Active player's king on the starting position, or an adjacent square if the king has castled. L3F0015: Enemy king in danger of being checkmated on the back rank. L3F0016: Square controlled by both friendly and enemy pawns. L3F0017: Enemy knight developed on the side of the board. L3F0018: Enemy knight attacked by active player's bishop and defended by enemy bishop. L3F0019: Not interpretable.

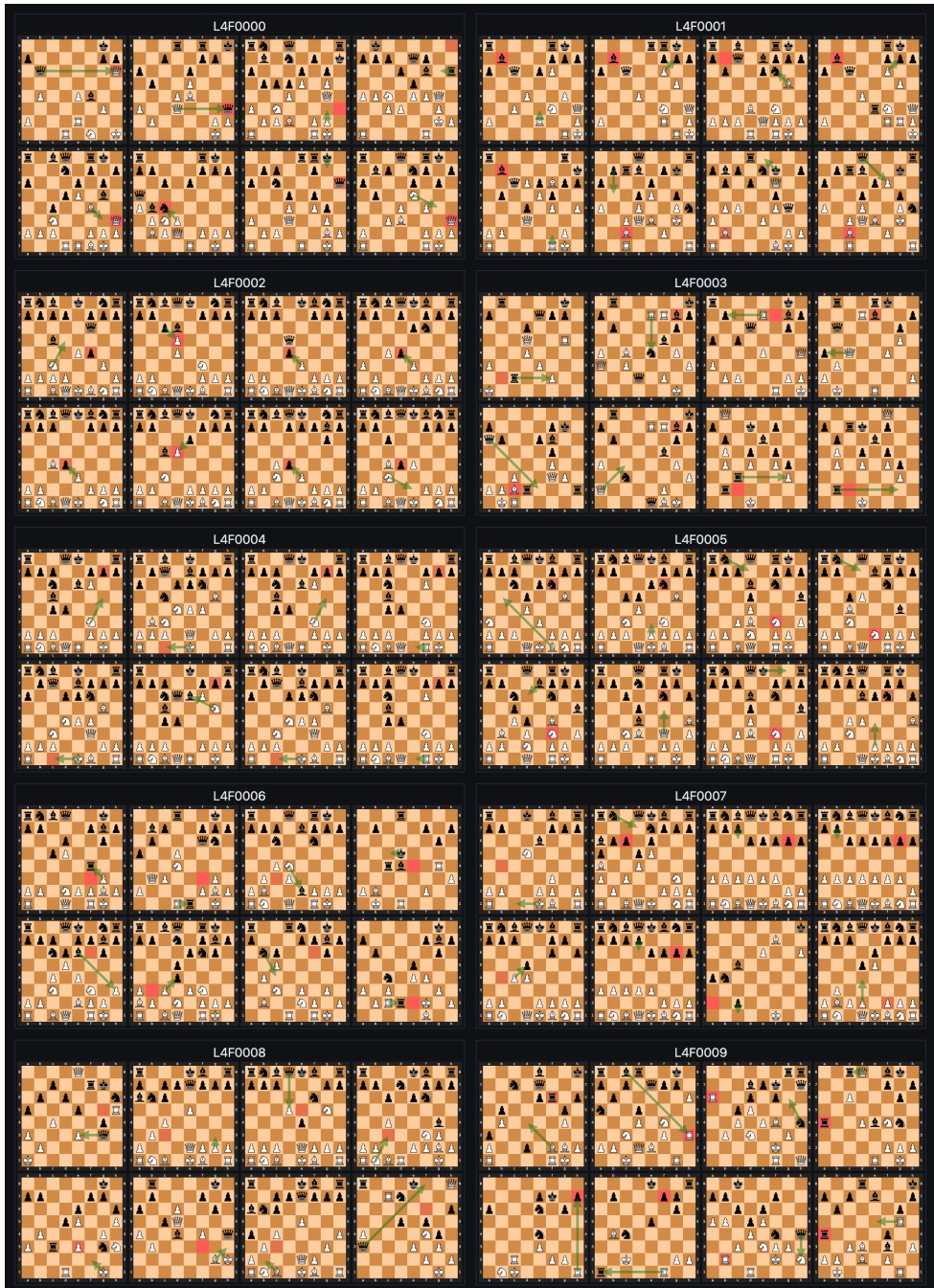

Figure 10: **Annotations for features 0-9 of layer 4 of** MAIA-3. L4F0000: Not interpretable. L4F0001: Active player's bishop on a strong diagonal, often paired up with a queen. L4F0002: Enemy center pawn targeted for capture in the opening. L4F0003: Square deep in opponent's territory attacked either by two rooks or a rook and a queen. L4F0004: Either long castling or tension between active player's f6 pawn and opponent's g7 pawn. L4F0005: Enemy knight pinned by the active player's bishop. L4F0006: Usually a square controlled by the active player's bishop. L4F0007: Not interpretable.

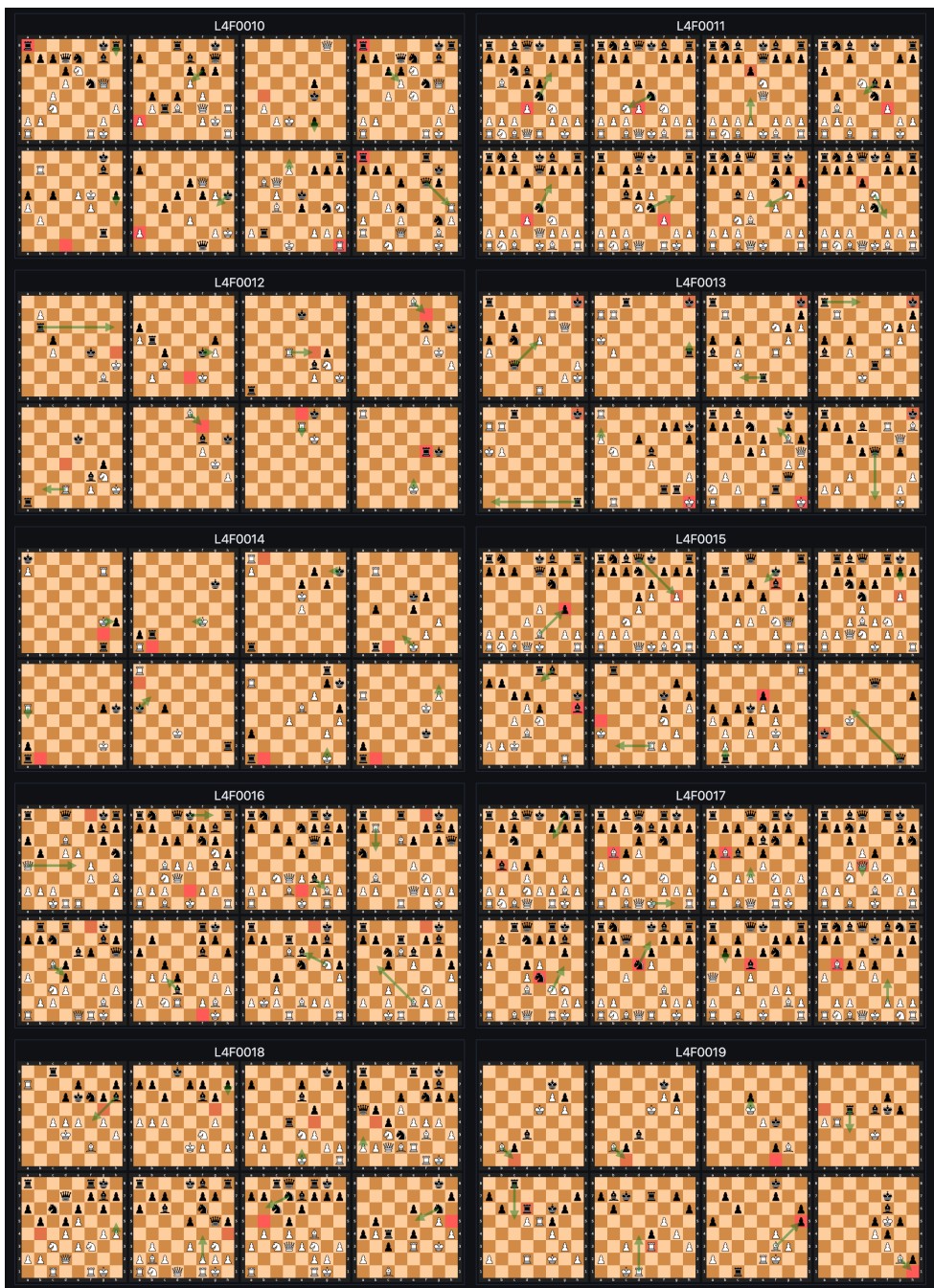

Figure 11: **Annotations for features 10-19 of layer 4 of** MAIA-3. L4F0010: Not interpretable. L4F0011: Enemy pawn attacking or threatening to attack an active player's minor piece. L4F0012: Not fully interpretable; miscellaneous key squares in endgames. L4F0013: Active player's vulnerable king in the corner. L4F0014: Square that is or will be controlled by enemy pawn, especially if it is close to promotion. L4F0015: Not interpretable. L4F0016: Square deep in opponent's territory controlled by active player's bishop. L4F0017: Active player's centralized piece in the middlegame. L4F0018: Key target square for enemy pawn push. L4F0019: Blockading square for enemy pawn.

## F.2 ADDITIONAL GAB AND ATTENTION HEAD HEAT MAPS

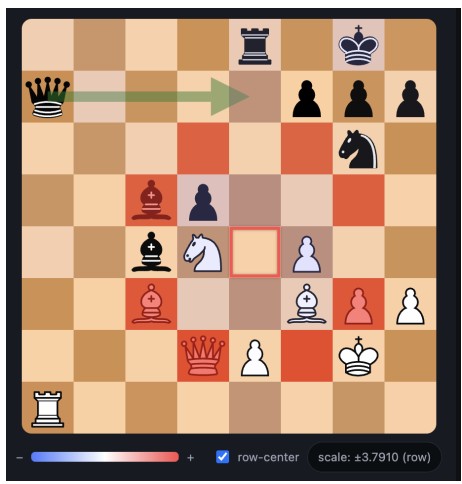 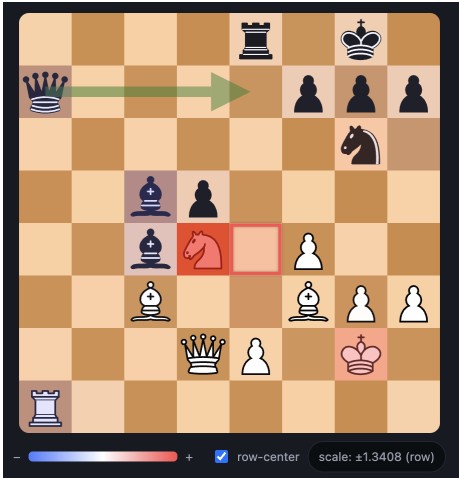

Figure 12: Layer 4 head 4 MAIA-3 GAB and DPA maps, left and right respectively.

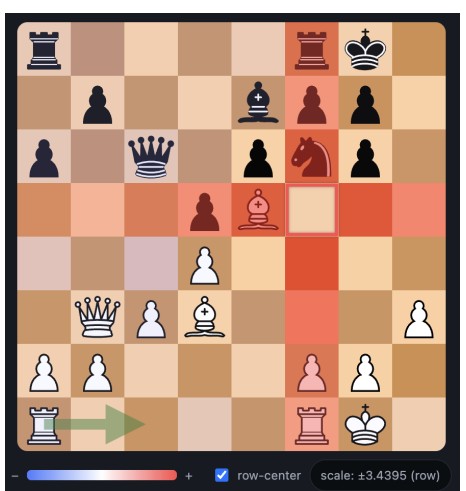 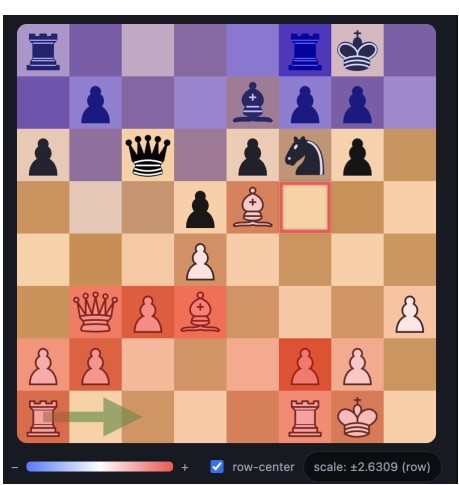

Figure 13: Layer 4 head 5 MAIA-3 GAB and DPA maps, left and right respectively.

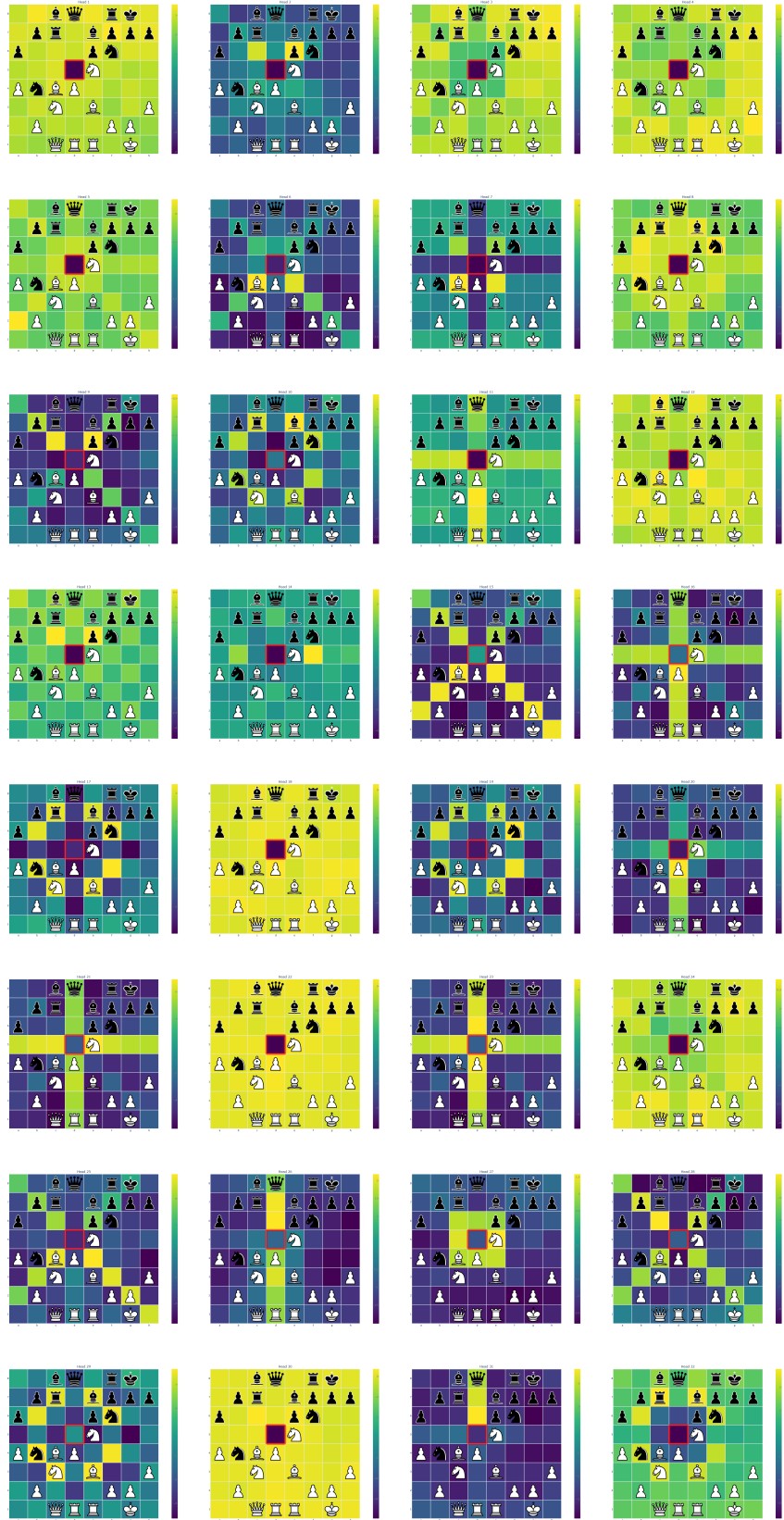

Figure 14: Additional Leela-CF GAB maps from layer 3.

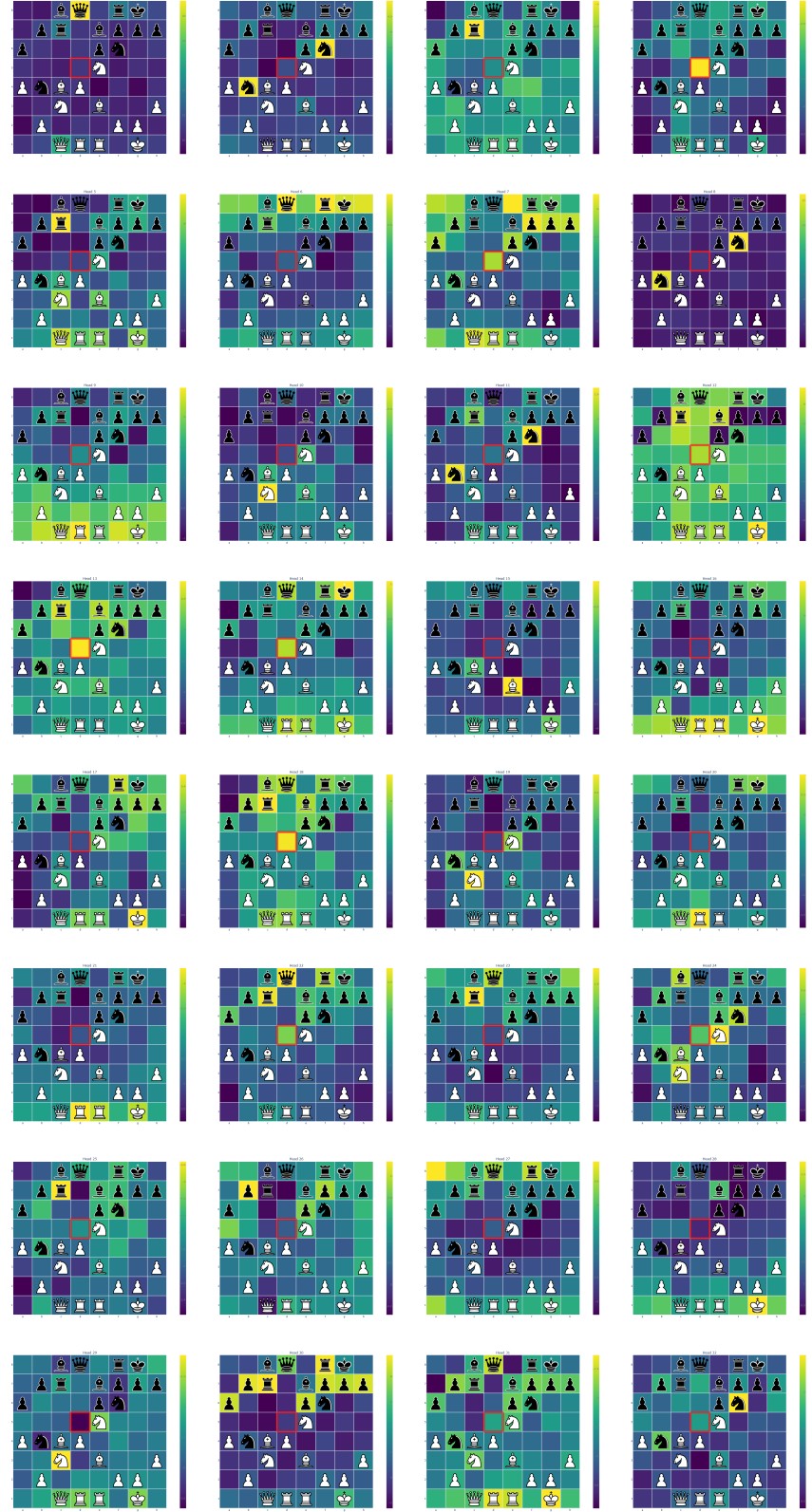

Figure 15: Additional Leela-CF DPA maps from layer 3.

