# OpenReview forum: "Chessformer: A Unified Architecture for Chess Modeling"
_ICLR.cc/2026/Conference — ICLR 2026 Poster_

### Official Review · Reviewer_6GBt · 2025-10-30

**Soundness:** 2
**Presentation:** 2
**Contribution:** 3
**Rating:** 6
**Confidence:** 4

**Summary:**

This work introduces a transformer architecture meant to be trained over next-move prediction in chess. The two main changes are (1) GAB and (2) formulating the policy prediction over moves more simply (from start to end square).

The authors take care to provide pseudonyms for models/works that are presumably related to their work. (Sometimes it makes it harder to understand the paper and the relative position of what they are trying contribute.)

**Strengths:**

* The writing itself is clear
* Figure 2 was interesting ; The tables 1,2 are clear
* Introduces a new intuitive way of framing how to format moves (an attention-based “source-destination” policy head)

**Weaknesses:**

* The first point raised in the abstract is only stated: "it significantly improves the chess-playing performance of a state-of-the-art chess engine." However, it does not describe this or give any information on this claim. While I am sympathetic to the challenge of anonymity issues, we cannot really take these 1-line statements into consideration.
* The first (and primary?) set of results looks at the loss and accuracy of the models vs. another model (Figure 1) vs. ablations on human scores (Table 1). The differences while real and positive are not really shown to be material. GAB exceeds the Absolute approach by 0.16 percent. Even if this were statistically significant it is not obvious it is meaningful. Likewise, the Figure 1 results are similar.
* Overall, it seems that the result is a strong one--a good architectural improvement--but that the demonstration of this based off of trust, not clear scientific ablations or clear documentation of what was done. This makes it hard to understand/evaluate the position of the paper.
* I did not find the interp. section compelling. There is a single figure comparison (cherry-picked? randomly-picked? representative?) in the first subsection. Likewise, in the next subsection they make similar one line note about SAE-like results. Again, it is not that I totally doubt the results are real. Rather, the results are insufficiently demonstrated.

**Questions:**

# Main Questions
* What is the primary goal of this work? It seems that the architecture is being sold as separate from the "state-of-the-art chess engine." When this is stated is this in reference to Table 3 (Table 3: Main results for raw playing strength.).
* The second listed contribution, the human-emulation matching, again, seems better, but marginally so, to the point it is unclear (or not sufficiently). The third, again, is the interpretability, and it seems again likely/possibly interesting, but just not shown.
* Do you report or comment on the stat. range or std of the scores reported in your tables? Anything to help contextualize the results would be helpful. The accuracy deltas being so small make it hard to appreciate the results. Are we near a ceiling of performance? Is the task hard or is the data stochastic?
* Q: "Empirically, we find that GAB is a key driver of these gains." Note that in Table 1, GAB also requires more FLOPS and param. Where is this clearly demonstrated?


# Minor Questions
* Q: In section 3.2, "we concatenate representations of the past 7..." How are the concat? I am reading this as "stacking" the information onto each of the 64 tokens. What are the final dimensions?
* Q: Are the embeddings for the weak/strong players learned/updated? (Any comments on the relatively large dimension for this 128dim; the rest of the embeddings is 12 for the pieces + some other auxiliary information?) It is unclear what the actual/final architecture and dimensions of the model are.

# Minor Notes
* Table 2. The parameter sizes are very different from those before. I think grouping together the different models and sizes actually used into a clear section would help.
* As shown in Table 5.3 --> As shown in Table 3,
* L429 "finder"
* L247 "achieves its largest gains at the highest calibers of strength *shows that* Chessformer" --> "suggests that"

---

> ### Author Response · Authors · 2025-11-26
> **Rebuttal**
>
> We thank the reviewer for their comments. We have made a number of changes that we in particular hope better contextualize our paper and demonstrate our results. However, please let us know if there are still any other sources of confusion, as clarity and reproducibility are two major goals of this work. We first summarize the changes that hopefully address a focal point of the review.
>
> **Clear scientific ablations and documentation**
>
> We appreciate that you found our results strong. We however agree that the initial submission could have benefited significantly from more deeply expounding on both results and methods. As this seems to be a focal concern of the review, we briefly review the changes we have made in this first revision that address it.
>
> ***Scientific rigor***
> We have added confidence intervals for all Elo and accuracy measurements. Because we found that the Allie test set does not have enough samples at low and high ratings to produce statistically significant results at those skill levels (for example, 205 positions above 2850 Elo), we have formed the Allie-Augmented test set to enable statistically significant results across Elo ranges. These per-Elo results are presented in Appendix E.
>
> ***Documentation***
> 	We have clarified how all test and training sets were made, as well as the full inputs for our models. To aid with reproducibility, we also promise to open-source our full pipeline,
> including all code and training data.
>
> ***Demonstration of results***
> We have justified our claims about improving engine strength by performing a tournament between configurations of Apollo equipped with a Chessformer model and the Apollo-CNN ResNet that Apollo had last used at a tournament. We have also expounded on three prominent chess engine tournaments won by Apollo configurations equipped with Chessformer models in Section 5.4.
>  To give a wider view of our interpretability results, we have included and interpreted 40 features from layers 3 and 4 of Zeus-79M in Appendix F (since feature ordering is arbitrary, these are effectively random).
>
> ***Depth of analysis***
> We have significantly updated the new revision to clarify our human emulation results. For that task, we have extended training runs to 1 million steps and trained models at scales of 5M, 23M and 70M, redoing our ablations at the smallest 5M-parameter to enable more rigorous comparisons with more fully trained models. We have also added Appendix E, which analyzes the effects of model scale, history information, and position encoding on performance by Elo, and then attempts to understand the position of our human emulation results in the context of the performance ceiling and stochasticity of the task.
>
> **Chess engine strength**
>
> We have added Section 5.4, which justifies our claims about significantly pushing the strength of a state-of-the-art chess engine. It compares the playing strengths of Apollo configurations paired with either the Apollo-CF Chessformer model or Apollo-CNN, the latest non-Chessformer used by Apollo at an official tournament. We see gains of 100 Elo points across time controls. For reference, under a similar testing setup, the top Stockfish engine gained around 46 Elo points in the 14 months between the releases of Stockfish versions 16 and 17 (Stockfish Team, 2024). The improvement is notable given not only how well-studied playing strength is in chess, but also the difficulty of improving top chess engines; to achieve these gains, Stockfish volunteers tested over 10,000 changes on thousands of CPU cores. We achieve a gain twice the size with a single architectural change.
> We have also added a high-level description of three events where Chessformer models, when integrated with the Apollo chess engine, won prominent chess engine tournaments against fields of competitors that included the Stockfish engine.

---

> ### Author Response · Authors · 2025-11-26
> **Rebuttal (Continued)**
>
> **Small improvements in human emulation**
>
> Crucially, on the human emulation task, we see a gain in parameter efficiency of over an order of magnitude, outperforming the 355M parameter state-of-the-art with a 23M Chessformer model and achieving comparable performance with a 5M Chessformer model. This shows that our methodology enables a substantial improvement in modeling efficiency.
>
> Despite these large improvements in efficiency, our models appear to be running into an accuracy ceiling. Even with the improvement we obtained since the first revision, we are advancing the move-matching accuracy of the state of the art by roughly 1.2±0.2%. There appears to be substantial accuracy compression on the human emulation task, reflected in the fact that increasing parameter count 14-fold from 5M to 79M only improves move-matching accuracy by 1.7±0.2% (55.4->57.1), matching the gain in policy accuracy when doubling parameter count on the Apollo distillation task (56.98±0.1 GAB-2.5M -> 58.53±0.1 GAB-4M).
>
> Figures 5 and 6, newly added in this revision, give insight into why this is the case. Accuracy and perplexity at low ratings barely budge across model scale and other ablations, suggesting that even weak models can model weak players well enough to reach an accuracy ceiling. That performance at low ratings has a low ceiling is not altogether too surprising, since weak play tends to be unsophisticated, inconsistent, and stochastic. In other words, it is easy to model the patterns that are present (reduce the epistemic uncertainty), but one quickly runs into a ceiling of around 50% imposed by the high aleatoric uncertainty. At high skill levels, on the other hand, play is consistent but maximally sophisticated and difficult to model. In other words, aleatoric uncertainty is low and epistemic uncertainty is high, so there is much more room for improvement. These high skill levels are where our models shine: as shown in Figure 5, Zeus-79M improves accuracy on 2800-rated play by over 5%.
>
> The outcome of human games (the 0.16% gain you referred to) is even more stochastic than the played move since it is effectively a sum of accumulated errors across several dozen moves. This metric has not typically been reported in prior work on human emulation and is not our primary interest, but we include it for completeness. Our main focus is on move-matching, and our techniques significantly expand on the state of the art here. As stated above, we improve move-matching for strong play by up to 5% against the prior searchless state-of-the-art, which is larger than the overall difference between the state of the art and the first work to explore this problem, Maia 1.
>
>
>
> **Statistical rigor**
>
> We have added 95% confidence intervals for all accuracies and Elo values reported in tables. For the per-Elo graphs in Appendix E on the Allie-Augmented test set, we have noted that `For all game ratings other than 2900, the margin of error is less than a percent
>
>  **Interpretability**
>
> For GAB interpretability, we performed a quantitative analysis to support our claims about the complementary roles of GAB and dot product attention in the model as well as GAB’s adaptation. We have moved this analysis from the appendix to the main text (Section 6.1) for increased visibility.
> Your criticism of our transcoder interpretability discussion is well-founded. To provide a fuller, more representative view of the transcoder features, we annotate the first 20 features of layers 3 and 4 of Zeus-79M in Appendix F (essentially random, because the order of features in a transcoder is arbitrary). If this larger pseudorandom sample of features does not sufficiently address your concerns, we would be willing to share the entire feature set through our feature visualization website (we have not released the website yet due to anonymity concerns, but these could likely be circumvented through changes to the site).
>
> **Pareto improvement with GAB.**
>
> To show that GAB provides a Pareto improvement in accuracy, parameters, and FLOPS for both the oracle and human emulation tasks, we train models with GAB that are around half the size of the model scale at which we perform position ablations. In Table 2, this is Zeus-3M, and in Table 6, this is GAB-2.5M. For human emulation, Zeus-3M matches the performance of the 4.6-million parameter relative biases and absolute position ablations, a reduction in parameters of 30% and in FLOPS of 40%. In raw playing strength, the GAB-2.5M model similarly matches the 3.7-million parameter absolute position bias.

---

> ### Author Response · Authors · 2025-11-26
> **Rebuttal (Continued 2)**
>
> **Primary goal**
>
> The “state-of-the-art chess engine” we are referring to is the (anonymized) AlphaZero-clone Apollo, which long predates this work. Our contribution is the Chessformer modeling approach, and Apollo is a standalone project that we substantially improve with our architecture. Our primary goal is to facilitate and accelerate research that uses chess as a model system, several impactful examples of which are provided in Section 2, related work. However, we hope this work also contributes broader lessons about how adapting position encodings and tokenization schemes to the structure of a domain is critical for performance.
>
> **Input concatenation**
>
> The positions are concatenated along the hidden dimension, so that the context length is fixed at 64. The input dimensions for all human emulation models are (64, 352), where 352 = 2 * 128 + 8 * 12. Auxiliary information needed for compatibility with the Apollo setup is included for the oracle replication but not human replication, as it was not found to help in either setting. We chose a size of 128 for consistency with Maia 2, though any positive value has the same effective representation capacity; we formulated this input as a vector to enable, e.g., individual behavior replication We acknowledge that the embeddings take up the majority of the inputs and have not ablated this choice, but we doubt that other choices would give results.
> For the oracle task, we start with this (64,96) but concatenate auxiliary information to better integrate with the Apollo codebase. This includes indicators for whether each of the current and last 8 positions are a repetition, 4 indicators for castling legality, an indicator for whether black is to move, the number of moves since the last capture or pawn move, and also a 0 and 1 which were originally intended to allow convolutional models to detect edges. The final input shape before the token embedding is (64, 112). We have updated the human emulation and oracle distillation sections with this information.
>
>
> **Learnability of player strength embeddings**
>
> The 2x128 embedding parameters are learnable and are trained along with the rest of the model. This was insufficiently documented in the text, which we have rectified in Section 4.2: `we compute an embedding ek
> for an Elo rating k as a linear interpolation between two learnable embeddings...`
>
> **Typos**
> Fixed, thank you.

---

### Official Review · Reviewer_FWRW · 2025-11-01

**Soundness:** 4
**Presentation:** 2
**Contribution:** 3
**Rating:** 8
**Confidence:** 4

**Summary:**

The paper introduces a novel transformer architecture specialized to playing chess (e.g. approximating optimal play, or for imitating human play). A key novel component is GAP ("geometric attention bias"), which essentially adds an additional component computed by an MLP to the pre-softmax attention scores, as a more dynamic alternative to traditional positional encodings. The new "Chessformer" architecture also uses a new type of policy head. Experiments for both optimal and human-imitating play show clear performance gains over other architectures, even at lower parameter counts and inference costs. The new architecture also has properties that make it easier to interpret, and the paper takes initial steps towards understanding the functions of different model components.

**Strengths:**

- The architecture contains clever ideas and is well-motivated for the domain of chess.
- The empirical results are very impressive, demonstrating clear gains over previous work and ablations, even when using over an order of magnitude less inference compute than prior work. The evaluation methodology is convincing.
- I feel that there are lessons to be learned beyond only chess. The paper is a great example showing that a simple off-the-shelf transformer baseline can be decisively beaten in non-language domains with a more specialized architecture. And the motivation for GAP could apply to other domains as well where positions/distances aren't well-described by a static approach.
- I found the SAE interpretability results (in appendix B.1) highly intriguing.

**Weaknesses:**

- While I think some of the lessons from this paper could generalize to other domains, the target audience may still be a bit narrow.
- The description of the architecture (in particular GAP and the policy head) could likely be made easier to follow with some figures showing those novel components

**Questions:**

1. Does GAP fully replace any positional encoding? If so that seems interesting/surprising, since if I understand correctly, GAP only directly affects attention scores, so MLPs and attention value vectors would not directly receive any positional information. Did you experiment with e.g. both GAP and an absolute positional encoding? And do you have guesses for why putting positional encodings directly into the residual stream of the transformer (rather than only attention patterns) isn't important for performance?
2. How cherry-picked are the two SAE features shown in the appendix? E.g. did you look at 100 features and these were the only ones this interpretable, or are half the features roughly this clean?

---

> ### Author Response · Authors · 2025-11-26
> **Rebuttal**
>
> We thank the reviewer for their thoughtful review. Our responses to the weaknesses and questions are below.
>
> **Generalization to other domains**
>
> Though we focus on chess, our main lesson is that adapting the tokenization schemes and position representations of a model to the domain is critical for performance. It is notable that a single domain-grounded can advance the state of the art across human emulation, raw engine strength, and interpretability given how well-studied chess is.
>
> However, even in itself chess as a model system is large and impactful. To name a few examples, it has found application in RL (Farebrother et al., 2024), creative generative AI (Feng et al., 2025), mechanistic interpretability (Jenner et al., 2024), and human-AI cooperation, (Hamade et al. 2024). We hope our techniques both convey this general lesson and catalyze the large body of work that studies more general problems in the model system of chess.
>
> **Figure**
>
> We have added Figure 1, which illustrates the function of GAB as a position encoding that splits attention maps into semantic and positional components.
>
> **Does GAB fully replace any positional encoding?**
>
> An early experiment on the Apollo setup showed a negligible change (<0.1% policy accuracy) from adding an absolute position bias on top of GAB. We have run an ablation on the human emulation setup to retest this, omitting history information to make sure that the model cannot use history to somehow infer absolute position information. The run with the absolute encoding started out slightly ahead (~0.3% at 100k steps), but the performance closed up to nothing by a million steps, suggesting that relative position information is indeed sufficient.
>
> **Why does inserting positional encodings into the residual stream work?**
>
> Several works in vision and language have had success by only inserting position information into the attention weights, like Rotary Position Encodings (RoPE) and relative biases, so introducing positional information into the residual stream does not seem to be necessary for strong performance. However, we have seen MLP features which always activate at the same location, providing evidence that Chessformers are able to infer absolute position information. It is therefore possible Chessformer models learn to infer absolute position information solely from GAB, which would obviate the need for introducing positional information into the residual stream.
>
> **Transcoder interpretability**
>
> We have found that these features are interpretable at a high rate (about half in layer 4 and even higher in layer 3). We annotate the first 20 features of layers 3 and 4 in Appendix F. See our general comment for more details.

---

> > ### Comment · Reviewer_FWRW · 2025-11-27
> >
> > Thank you for the new figure, the interpretability results, and the explanation on GAB! I'm increasing my presentation sub-score since I think the nice new figure 1 helps quite a bit.
> >
> > As in my initial review, I do agree that the paper has relevance beyond chess. I think this paper would be a good contribution to the conference and will keep my already high overall score.

---

> > > ### Author Response · Authors · 2025-11-27
> > >
> > > Thank you for your quick response and positive feedback on our changes!

---

### Official Review · Reviewer_XrJD · 2025-11-01

**Soundness:** 4
**Presentation:** 2
**Contribution:** 3
**Rating:** 6
**Confidence:** 4

**Summary:**

This paper presents Chessformer, a transformer-based model for chess that unifies engine play, human move prediction, and interpretability on board attention. The model represents chess positions using 64 square tokens and introduces a Geometric Attention Bias (GAB), which is a dynamic positional encoding that adapts to board geometry. It also uses an attention-based policy head aligned with chess’s “from–to” move structure. Experiments show Chessformer outperforms prior models like Allie in both playing strength and human move prediction while being more efficient and interpretable.

**Strengths:**

- The introduction of GAB is a creative and well-motivated innovation that aligns the model’s attention with the spatial and semantic structure of the chessboard.
- The experiments are very comprehensive, covering both engine and human benchmarks. The ablation studies also show consistent performance gains.

**Weaknesses:**

- While the Geometric Attention Bias is central to the paper’s contribution, it is only presented in pseudocode within the appendix. A main-text figure illustrating its structure, input–output flow, and how it modulates attention across the board would make the concept far more intuitive and strengthen readers’ understanding.
- The description of how Lichess data were sampled and balanced across Elo levels is brief and lacks specific counts or sampling ratios, which may limit reproducibility.

**Questions:**

- How was the Lichess 2023 dataset processed in practice? How many samples per Elo range were used, and what criteria guided the downsampling to balance skill levels?
- You mentioned "Chessformer models mainly adapt the GAB biases to global positional features like the game stage (opening, middlegame, endgame), rather than the locations of individual pieces." Why is GAB able to recognize different game stages?

---

> ### Author Response · Authors · 2025-11-26
> **Rebuttal**
>
> We are grateful to the reviewer for their careful recommendations. Our responses to the weaknesses and questions follow:
>
>
> **GAB figure**
>
> We have added Figure 1, which shows the full updated attention mechanism and a strong example of how GAB collaborates with dot-product attention in practice. With the highlighted white rook on c1 as the querying square, the dot-product attention logits identify several important squares, while the GAB biases emphasize squares that are a rook’s move away. Together, they point the c1 rook to black’s weak knight on c5.
>
> **Dataset sampling**
>
> We split the dataset into chunks of 20,000 games, and split the Elo range into 20 buckets of size 100 from 600 to 2600, as well as two more buckets for the endpoints. During training we iterate through the games, calculate the game rating as the average of the two player ratings, and assign the game to the corresponding bin so long as it has at most 10 games. When each bin has 10 games or the chunk is exhausted, the training data is returned. Then from each game we uniformly sample 32 positions per game, or take every position if that many are
> not available. We have clarified this in Section 4.1 and Appendix A.1. On the topic of reproducibility, we promise to release our full pipeline, including all code and training data.
>
> **Recognition of games stages in GAB**
>
> Game stages usually correspond to the number of pieces left on the board; Lichess, for example, classifies endgames as positions where the major pieces (bishops, knights, rooks, and kings) number at most 6. A potential mechanism is that the first linear projection has units that activate when a certain piece is present at a square, and the next linear projection after the flattening layer counts the number of pieces of that type. This fundamentally aligns with our motivation for GAB as a position representation that can adapt to a position.

---

### Official Review · Reviewer_kykM · 2025-11-05

**Soundness:** 4
**Presentation:** 3
**Contribution:** 2
**Rating:** 4
**Confidence:** 5

**Summary:**

The paper introduces Chessformer, a novel transformer architecture specifically designed for the domain of chess, which significantly improves move prediction and playing strength over prior approaches. Chessformer makes multiple domain-specific architectural improvements: encoding the 64 board squares as tokens, using an attention-based “source-destination” policy head instead of naively one-hot encoding all possible legal moves, and using a Geoemtric Attention Bias (GAB) to convey positional information. Evaluations on move prediction and game-playing show that the Chessformer matches or outperforms prior approaches at a fraction of the cost.

**Strengths:**

The proposed approach is sensible and appears to work well in practice. The empirical validation is comprehensive and shows that Chessformers match or outperform prior work at a fraction of the scale and cost. The paper conducts ablations to show that the Geometric Attention Bias outperforms traditional positional encoding schemes. The Geometric Attention Bias and the “source-destination” attention head are novel and well-suited to chess. The proposed architecture modifications promise to facilitate domain-specific interpretability research by being more suited to the geometry of chess. The paper is well-written and easy to follow.

**Weaknesses:**

The main weakness of this work is that it is restricted to chess and, therefore, likely to be of marginal importance beyond the chess-ML community. Chess has served as an important testbed for many ideas in AI; however, this paper’s contribution is to make chess-specific adaptations to Ruoss et al. (2024) to obtain better performance. While the paper does a fine job at that, it is not quite clear to me what anyone outside of the chess community can learn from this work.

Given the above, the primary contribution of this paper should be to advance the state-of-the-art in the narrow subdomain of searchless chess modeling in a _reproducible_ manner, i.e., by releasing the code, model parameters, and/or the dataset. To the best of my knowledge, the paper does not address any of these aspects, unlike prior work (Ruoss et al., 2024).

The paper makes the unsubstantiated claim that “Chessformer significantly improves the chess-playing performance of a state-of-the-art chess engine” and “contributed to match wins over Stockfish in multiple computer-chess tournaments”. However, there is no empirical evidence to back up this claim.

The paper claims to use a novel encoding scheme; however, it is quite similar to the one proposed by Ruoss et al. (2024), who utilize an expanded FEN notation, i.e., 64 board states (and some additional information, which this paper also encodes). The main difference between the two approaches is that Ruoss et al. (2024) only feed the current board state to the transformer, while this paper proposes to concatenate it with the previous 7 board states.

There are a few typos:
* L269 “name changed”
* L352 “absolute position”
* L355 “Table 3”
* L376 “A recent line”
* L429 “much finer interpretation”

**Questions:**

* How did the paper arrive at concatenating the current and the 7 past positions? It would be interesting to ablate this particular architectural choice.

---

> ### Author Response · Authors · 2025-11-26
> **Rebuttal**
>
> We are grateful to the reviewer for their comments and recommendations. Our responses are as follows:
>
>
> **Reproducibility**
>
> We concur that reproducibility will be essential to the lasting impact of this paper. Our main goal is to provide a “Swiss-army architecture” for chess-modeling problems, just as the Ruoss paper’s well-documented codebase has allowed it to serve as a “Swiss-army training setup” for research using chess as a testbed, e.g., Farebrother et al. (2024) and Feng et al. (2025). As a reviewer for that paper mentioned, projects such as Leela Chess Zero are poorly documented and therefore of limited value to the scientific community. With this in mind, we commit to making this project reproducible. We have added this sentence to lines 66-67 in the introduction: `We open-source our full pipeline, including all code and training data, at (link removed for anonymity).`
>
> **Generalization beyond chess**
>
> The general lesson is that adapting a model’s tokenization and position representation schemes can greatly improve performance. It is surprising that a domain-aligned architecture can simultaneously advance human chess modeling, raw engine strength, and interpretability by a significant margin, especially given that both chess-playing strength and human emulation are competitive problems, and that past approaches have focused on only one of these at a time.
>
> Our work has the potential to catalyze future work in the chess-AI community, which we emphasize is large and impactful. To name a few works, Farebrother et al. (2024) demonstrated through chess that classifying rather than regressing improves scalability of RL, and Feng et al. (2025) proposed methods to enhance the creativity of generative AI through chess puzzles. McIlroy-Young et al. (2021) and McIlroy-Young et al. (2022) explored individual behavior stylometry, and Hamade et al. (2024) later investigated human-AI cooperation in the testbed of chess.
>
> These papers all rely heavily on prior chess modeling techniques, and an open-source, reproducible architecture that is an order of magnitude more parameter and compute efficient than existing techniques would greatly accelerate this type of research, both reducing training costs and enabling much more intelligent modeling. Our Zeus-79M model, for example, achieved state-of-the-art (56% accuracy) move-matching accuracy after 24 hours, outperforming Allie methods that took 2 weeks to train on comparable hardware. We hope that by open-sourcing our work, we will not only contribute lessons about general modeling principles but also catalyse the large body of research that uses chess as a testbed.
>
>
> **Engine strength and tournament performance**
>
> We acknowledge that our claims about engine strength and tournament performance were not justified and have attempted to rectify this through additions to Section 5.4 which addresses them both. In particular, it describes the format, models used, and results of three prominent online chess engine tournaments where Chessformer-equipped Apollo configurations defeated large pools of engines that included the top Stockfish engine. It also compares the playing strength of an Apollo configuration equipped with the Apollo-CF Chessformer model against the older convolution-based Apollo-CNN model that Apollo had used in the most recent major tournament.  We see consistent gains of 100 Elo points, which is especially large for a top engine. For context, in the 14 months between June 2023 and September 2024, the Stockfish project continuously employed over a thousand CPU cores to run over 10 thousand volunteer-submitted tests, gaining roughly 46 Elo on a similar testing setup.

---

> ### Author Response · Authors · 2025-11-26
> **Rebuttal (Continued)**
>
> **Tokenization scheme**
>
> We acknowledge that some of our wording seemed to imply that the tokenization was fundamentally different when the main difference between our work and Ruoss et al. is the position representation. We have clarified this in the new revision, adding more detailed architectural comparisons in Section 3.1 and Appendix D. For example, we no longer claim that Ruoss et al. used `bespoke notational tokenization that are potentially mis-aligned with the underlying action space`
> However, though the 64-square representation has been used before by Ruoss et al. (2024), our novelty is marrying this 64-token representation with a position encoding that flexibly and dynamically reflects the underlying board structure. Ruoss et al. used rotary position embeddings, modeling positional relationships in a one-dimensional way that is not grounded in chess principles. As we write in the new revision, `For example, among relationships between squares, [Ruoss’s] architecture maximally decays the attention strength between opposite corners, even though those corners lie on a main diagonal that is critical for checkmate patterns.`
>
> The goal of a position encoding should be to allow the QK component of attention to focus on semantic relationships. Because GAB is dynamic and learnable, it flexibly adapts to the variable positional relationships of chess. This difference in formulation is conceptually simple but empirically gives enormous efficiency games. Another small but notable difference is the exclusion of tokens for auxiliary information. While we use 64 tokens, Ruoss et al. used 77. Our design is therefore both slightly faster and also more interpretable, since it prevents information from flowing off-board into “register tokens” that cannot be interpreted with respect to a given square.
>
> Our new revision focuses on this key difference when comparing to the Ruoss paper. In particular, we say:
>  ```Our setup is most similar to that of Ruoss et al. (2024), which tokenized the 64 board squares in addition to other information about the position, for a total of 77 tokens. However, that work adopted rotary position embeddings (Su et al., 2024) on a linearized representation of the board squares, enforcing a one-dimensional structure on the board that is not grounded in the domain…```
>
> **Position History**
>
> Initial experiments with a 79M model on the human move-matching task showed a 1.5% accuracy gain from increasing the amount of history information from no past positions to the past seven positions, and no additional gain from increasing to the past 31 positions. We have redone this experiment under our final ablation setup at the 5M scale and are seeing nearly identical results. In other words, 7 history positions seems like a good middle ground. These results have been added in Figures 6 and 7 in Appendix E. Interestingly, the gains appear to be largest for weak play, which we suspect is due to the higher initial aleatoric uncertainty for those players.
>
> **Typos**
> Fixed.

---

### Author Response · Authors · 2025-11-26
**Summary of Changes**

We are grateful for the thoughtful comments and feedback provided by the reviewers. We are pleased that the reviewers appreciated the novelty and well-suitedness of our techniques (all reviewers), as well as the clarity of our writing (Reviewers kykM and 6GBt) and the convincingness (Reviewer FWRW) and thoroughness (Reviewer XrJD) of our evaluations.

To make the revisions clearer, we highlight in blue all text that contains fundamentally new or different information. We first describe general changes made to the paper, then discuss our responses to reviewer suggestions.

**New Results**

We have significantly updated our results on the human move-matching task to include models at scales of 5M, 23M, and 79M (the model configuration for 79M is the same as in the initial submission; we are now rounding to the nearest million rather than taking floors). *In brief, we now outperform the previous state of the art (Allie) with 10x fewer parameters, and achieve comparable performance with 70x fewer parameters.* We standardized the training setup and extended the length of training runs from 200K to 1M steps, training on data from January 2023 to July 2025 rather than January 2023 to December 2023. This increased move-matching accuracy at the 79M scale from 56.6% to 57.1% and enabled our 23M model to match the original 56.6% performance of the 79M model. We have redone the ablations for human emulation at the 5M scale and are getting results close to those of the reduced-compute 79M ablations, suggesting that our initial training runs were not long enough. At the 5M scale, we obtain a move-matching accuracy of 55.4%.

**Figure** (FWRW and XrJD)

Two reviewers made the helpful suggestion to add an explanatory figure. We have now added main-text Figure 1, which illustrates the function of the GAB encoding, and in particular the way it decomposes the attention computation into semantic and positional components. The GAB biases model piece movement, freeing up the dot-product component to focus on finer semantic information. Positional relationships modeled by GAB are both flexible and domain-grounded, enabling a much more efficient attention layer than prior work that simply enforced a one-dimensional structure on the squares (Ruoss et al., 2024). We feel this is a very helpful addition and thank the reviewers for this suggestion.

---

> ### Author Response · Authors · 2025-11-26
> **Summary of Changes (Continued)**
>
> **Broader applicability** (kykM, FWRW)
>
> Concerns were brought up about the narrowness of the paper’s audience. We appreciate this feedback and have clarified the motivation and implications of our work, so that its relevance is better communicated to readers.
>
>
> First, we underscore that chess is widely used as a model system. In reinforcement learning, Farebrother et al. (2024) used chess to demonstrate that classifying rather than regressing improves scalability of RL, and Feng et al. (2025) proposed methods to enhance creativity in generative AI in the testbed of chess puzzles. Hamade et al. (2024) investigated human-AI cooperation in the testbed of chess and McIlroy-Young et al. (2021) and McIlroy-Young et al. (2022) explored individual behavior stylometry. Critically, all these works relied heavily on prior work like Ruoss et al. (2024) or Maia 2 (Tang et al., 2024) for their models. Having an open-source, general chess architecture that improves parameter and compute efficiency by over an order of magnitude across modeling tasks would greatly accelerate this line of work. Our Zeus-79M model, for example, achieves 56% move-matching accuracy on the Allie test set in 24 hours, outperforming state-of-the-art Allie methods that were trained for 2 weeks on similar hardware.
>
> Second, we emphasize our central lesson about the importance of domain-aligned tokenization and position representation schemes. We believe it will come as a surprise to the community that a single, well-designed architecture can simultaneously enable state-of-the-art models for the very different goals of maximizing raw chess-playing strength (a very competitive problem that goes back to Deep Blue and has been reinvigorated with the development of deep learning based approaches like AlphaZero), faithful human behaviour modeling (with relatively mature models such as Maia and Allie), and interpretable decision-making (making it more transparently clear what the models are attending to). Currently, the strongest chess engines in the world use a mixture of classical search methods and shallow neural networks, while the best human move predictors use ResNets (Maia) or standard Transformer architectures (Allie), neither of which is particularly interpretable. It is notable that a single architecture such as Chessformer can simultaneously achieve state-of-the-art performance on these disparate problems, which have until now been optimized with bespoke, but suboptimal, approaches. The general lesson here is that substantial, general progress can still be made with careful architecture designs that respect and are adapted to the domain(s) at hand, even in simple model systems, and furthermore even in model systems that have been amply studied, such as chess.
>
> **Interpretability** (6GBt, FWRW)
>
> Multiple reviewers found our discussion of transcoder feature interpretability interesting, but concerns were raised about the representativeness of the features included in the paper. To provide the reviewers with a fuller view of interpretable features identified in Zeus, we retrain transcoders on the slightly stronger Zeus checkpoint reported in this revision and annotate the first 20 features of layers 3 and 4 in the appendix (the order of features in the transcoder is arbitrary, so these features are essentially random). We train on layers 3 and 4 because (a) of compute budget limitations imposed by other ablations we had to run and b) our preliminary investigations with previous transcoders found that these layers contained the most interpretable representations. Training an SAE or transcoder on just the middle layers of a model is standard practice, as middle layers often contain the most interpretable representations (Gurnee et al., 2023). [Appendix F] If this larger random sample of features does not sufficiently address the reviewers’ concerns, we can share the entire feature set by anonymizing our feature visualization website, so that reviewers can go through the features and judge their interpretability for themselves. The important takeaways are that many interpretable features can be identified in our model, and that a large share of these features are only interpretable (or more interpretable) by virtue of the fact that our architecture allows for the specific squares which activate features the most to be examined. [Section 6.2]
>
> Additionally, we have moved the quantitative analysis of how GAB and dot-product attention (DPA) vary between and within positions to the main text (Table 4) in response to reviewer 6GBt’s concerns about our GAP interpretability results. Our findings across 30,000 positions provide evidence for the complementary roles of GAB and DPA as encoders of positional and semantic information respectively. We also find that GAB biases for a given query square vary significantly across positions, supporting our motivation for GAB as a positional encoding system that can adapt to different positions.

---

> ### Author Response · Authors · 2025-11-26
> **Summary of Changes (Continued 2)**
>
> **Chess engine performance** (kykM, 6GBt)
>
> We claimed that our proposed architecture was integrated into configurations of a top engine, greatly increasing its playing strength and allowing it to defeat the top Stockfish engine at several prominent computer chess tournaments. We had to balance sharing results with anonymity concerns and are still balancing these, though we acknowledge that our initial submission did not give the reviewers enough evidence. We have therefore added analysis to Section 5.4, which (1) estimates the Elo gain from swapping in the Apollo-CF model for the Apollo-CNN model that Apollo previously used at top tournaments and (2) describes both the high-level format and results of events where Chessformer-equipped Apollo configurations defeated Stockfish. We have also added more concrete details on our Apollo-CF vs Apollo-CNN tournament setup to Appendix B.

---

### Author Response · Authors · 2025-12-03
**Summary of Discussion for Area Chair**

Dear area chair,

We would like to thank the reviewers for helping us significantly improve our manuscript, especially in light of the unfortunate circumstances. To summarize our changes, we have justified claims about improving engine strength, trained models at a wider variety of scales for the human emulation task, and added more details and analysis.

**New Results**

We have significantly updated our results on the human move-matching task to include models at scales of 5M, 23M, and 79M (the model configuration for 79M is the same as in the initial submission; we are now rounding to the nearest million rather than taking floors). **In brief, we now outperform the previous state of the art (Allie) with 10x fewer parameters, and achieve comparable performance with 70x fewer parameters.** We standardized the training setup and extended the length of training runs from 200K to 1M steps, training on data from January 2023 to July 2025 rather than January 2023 to December 2023. This increased move-matching accuracy at the 79M scale from 56.6% to 57.1% and enabled our 23M model to match the original 56.6% performance of the 79M model. We have redone the ablations for human emulation at the 5M scale and are getting results close to those of the reduced-compute 79M ablations. At the 5M scale, we obtain a move-matching accuracy of 55.4%.

**Presentation**

Following the suggestions of Reviewers FWRW and XrJD, we have added a main-text figure, Figure 1, illustrating the function of our proposed position representation.

Reviewer kykM stated that we had seemed to mischaracterize the tokenization scheme used by Ruoss et al. (2024). We acknowledge that this was initially unclear, and have reworded the relevant sections to emphasize that our main difference is the positional representation that the tokenization is married with.

**Documentation**

Reviewer kykM expressed that our work should be reproducible, which we had originally failed to document. To rectify this, we have added an assurance that we will release all code and data.

We have addressed concerns raised by Reviewers kykM and 6GBt about insufficiently justified claims by performing an engine tournament between Apollo configurations equipped either with the Apollo-CF Chessformer or the Apollo-CNN convolution model previously used at tournaments. We have also added high-level descriptions of three events won by Chessformer-equipped Apollo configurations against fields of competitors that included the top Stockfish engine.

Following concerns of Reviewers 6GBt and FWRW, we have strengthened the interpretability section by adding 40 arbitrarily chosen transcoder features to the Appendix, and by moving a previously-buried Appendix analysis into the main text which quantitatively demonstrates the complementary, interpretable roles that GAB and DPA play.

Reviewer XrJD believed our training filtering process should have been better documented. We have clarified the exact methodology in Section 4.1 and Appendix A.1.

Reviewer 6GBt had broad concerns about the demonstration of our results. To address these, we have added 95% confidence intervals for all accuracies and Elo values and explicitly noted that GAB beats other positional encodings with fewer FLOPS and parameters to achieve a Pareto improvement. We also added Appendix E, which aims to understand the small increase in performance metrics based on the aleatoric and epistemic uncertainty of the task.

Reviewer kykM was interested in our choice of concatenating the past 7 board states. We have done an ablation on the human modeling setup that is reported in Appendix E. As shown in Figures 5 and 6, we say that history information helps more at lower game ratings.


**Motivation**

Reviewers kykM and FWRW raised concerns about the potential size of the audience for our work. We reiterate our response to these concerns here:

Our work comes at a time when chess is widely used as a model system in reinforcement learning (Feng et al. (2025) and Farebrother et al. (2024)), human-AI interaction (Hamade et al. (2024)), behavioral stylometry (McIlroy-Young et al. (2021) and McIlroy-Young et al. (2022)), and mechanistic interpretability (Karvonen et al. (2024) and Jenner et al. (2024)). All these works depend on prior modeling approaches, and our architectural improvements have the potential to greatly accelerate chess related research across the board by reducing compute costs and enabling more capable and interpretable models.

We further emphasize that our proposed architecture not only has the capacity to catalyze the wide range of work that uses chess as a model system, but we also believe it will be a valuable example for the broader ML community that aligning modeling form with action space can lead to such large performance gains across a variety of top-level tasks, even in domains as well-studied as chess.


**Conclusion**

We again thank the reviewers for assisting us in improving the manuscript.

---

### Meta-Review · Area_Chair_iu7B · 2026-01-10

**Summary:**

This paper introduces Chessformer, a transformer architecture tailored to chess that aims to unify three goals: strong engine play, accurate human move prediction across skill levels, and improved interpretability. The key design choices include representing chess positions as 64 square tokens, introducing a Geometric Attention Bias (GAB) as a dynamic, domain-aligned positional encoding, and using an attention-based “source–destination” policy head aligned with the chess move structure. The paper presents extensive empirical evaluations on both engine-strength benchmarks and human move-matching tasks, showing that Chessformer matches or outperforms prior approaches with substantially fewer parameters and lower compute, while also enabling more interpretable internal representations.

Overall, reviewers agreed that this is a technically solid and well-motivated architecture paper with impressive empirical results. The main concerns raised focused on the breadth of applicability beyond chess, the clarity and documentation of some architectural and experimental details, the strength of evidence supporting claims about engine performance, and the representativeness of the interpretability analyses. These concerns primarily affected presentation, positioning, and reproducibility, rather than the core technical contribution.

**Reviewer Concerns:**

The main concerns are: (1) generalization beyond chess, (2) documentation for reproducibility, and (3) some claims lacking empirical support.

For (1), while I do not think this concern is fully resolved, I agree with the authors and several reviewers that chess has long been used as a model system for a wide range of studies, and that meaningful improvements within this domain alone constitute a valuable contribution.

For (2), the authors have clarified the details raised in the reviews and have committed to open-sourcing the code and data. While the response would have been more convincing if an anonymized repository had been provided, I am inclined to trust the authors to follow through on this commitment, particularly given the accountability afforded by the open nature of OpenReview.

For (3), the authors have added additional experiments and updated their claims, which largely alleviates the reviewers’ concerns.

**Reviewer Scores:**

Three of the reviewers were already positive about the paper prior to the rebuttal.

Reviewer kykM gave an initial score of 4, noting the three concerns above. If the authors had provided an anonymized repository, I expect this reviewer would have raised the score to around 6. While the authors only promise to do so, I am inclined to trust them on this point.

Overall, I recommend acceptance, based on the expectation that the authors will open-source their code and data as promised.

---

### Decision · Program_Chairs · 2026-01-26

Accept (Poster)